# Offline Preference Optimization for Rectified Flow with Noise-Tracked Pairs

**Yunhong Lu** [1]  **Qichao Wang** [1]  **Hengyuan Cao** [1]  **Xiaoyin Xu** [1]  **Min Zhang** [1 2 3]

## Abstract

Existing preference datasets for text-to-image models typically store only the final winner/loser images. This representation is insufficient for rectified flow (RF) models, whose generation is naturally indexed by a specific prior noise sample and follows a nearly straight denoising trajectory. In contrast, prior DPO-style alignment for diffusion models commonly estimates trajectories using an independent forward noising process, which can be mismatched to the true reverse dynamics and introduces unnecessary variance. We propose Prior Noise-Aware Preference Optimization (PNAPO), an off-policy alignment framework specialized for rectified flow. PNAPO augments preference data by retaining the paired prior noises used to generate each winner/loser image, turning the standard (prompt, winner, loser) triplet into a sextuple. Leveraging the straight-line property of RF, we estimate intermediate states via noise-image interpolation, which constrains the trajectory estimation space and yields a tighter surrogate objective for preference optimization. In addition, we introduce a dynamic regularization strategy that adapts the DPO regularization based on (i) the reward gap between winner and loser and (ii) training progress, improving stability and sample efficiency. Experiments on state-of-the-art RF T2I backbones show that PNAPO consistently improves preference metrics while substantially reducing training compute.

## 1. Introduction

Text-to-image generation has progressed rapidly with diffusion models (Rombach et al., 2021; Podell et al., 2023)

[1]Zhejiang University [2]Shanghai Institute for Advanced Study-Zhejiang University [3]Shanghai Institute for Mathematics and Interdisciplinary Sciences. Correspondence to: Min Zhang <min_zhang@zju.edu.cn>.

*Proceedings of the 43rd International Conference on Machine Learning*, Seoul, South Korea. PMLR 306, 2026. Copyright 2026 by the author(s).

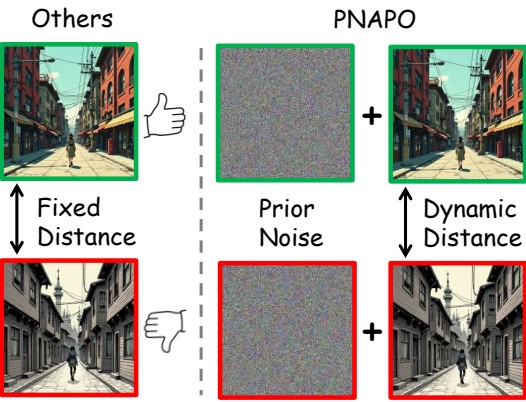

Others  PNAPO

Fixed Distance · Prior Noise · Dynamic Distance

Steampunk city street with people by junji ito trending on artstation

*Figure 1.* Our PNAPO achieves self-improvement by utilizing prior noise distributions and dynamically adjusting gradient updates.

and, more recently, rectified flow (Esser et al., 2024) and flow-matching (Lipman et al., 2022) variants. Despite their success, high-capacity T2I models still exhibit persistent failure modes: imperfect text rendering (Chen et al., 2023), compositional errors (Huang et al., 2023), spatial inconsistencies (Lin et al., 2024), and hallucinated objects (Ren et al., 2023). Many remedies (scaling data (Gadre et al., 2023), retraining from scratch (Karras et al., 2022), architecture changes (Peebles & Xie, 2022; Pernias et al., 2023), or adding semantic conditioning (Chen et al., 2024)) are costly and often orthogonal to what users ultimately want: *human-preferred* outputs. This motivates post-training alignment via preferences, analogous in spirit to reinforcement learning from human feedback (RLHF) (Ouyang et al., 2022).

A standard preference optimization pipeline for T2I models has two stages: (i) collect preference pairs for prompts, and (ii) optimize the generator to increase the likelihood of winners relative to losers, typically using reward models (Clark et al., 2023; Prabhudesai et al., 2023), RL objectives (Black et al., 2023; Fan et al., 2023; Zhang et al., 2024b), or RL-free DPO-style (Wallace et al., 2024) surrogates. While RL-free methods are attractive due to stability and simplicity, a central issue is frequently glossed over: **preference datasets usually store only final images** (Kirstain et al., 2023; Lee et al., 2023; Liang et al., 2024; Wu et al., 2023; Zhang et al., 2024a). For diffusion-like models, however,

## Text Alignment

## Aesthetics and Realism

## Clear Background

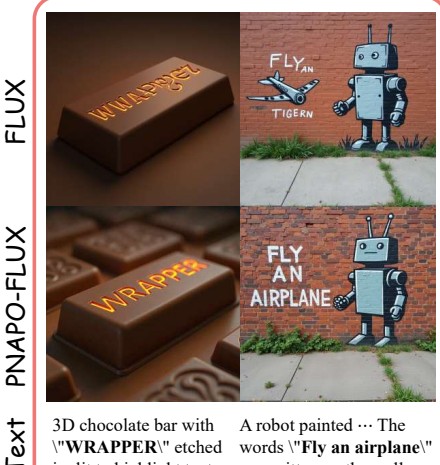
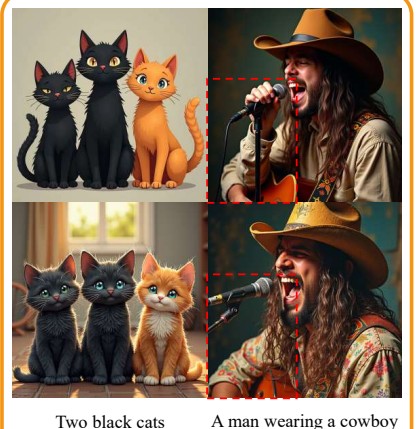
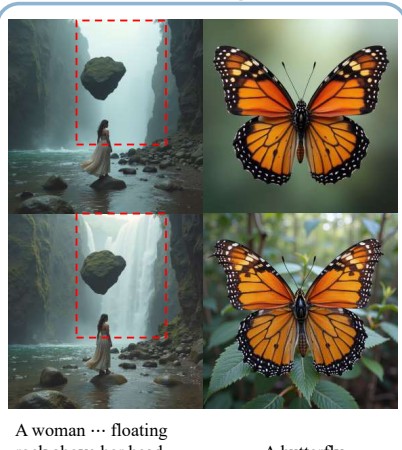

FLUX

PNAPO-FLUX

Text

3D chocolate bar with \"**WRAPPER**\" etched in, lit to highlight text.

A robot painted ⋯ The words \"**Fly an airplane**\" are written on the wall.

Two black cats standing next to an orange cat.

A man wearing a cowboy hat and screaming a punk song into a microphone.

A woman ⋯ floating rock above her head ⋯ cascading **waterfall**.

A butterfly.

*Figure 2.* **Prior Noise Matters.** Compared to FLUX, our `PNAPO`-FLUX generates images with superior text-image alignment, enhanced visual aesthetics and realism, particularly in resolving FLUX's characteristic background blurring issues. These advancements parallel how LLMs address hallucination, as both represent implicit optimizations of human preference alignment.

the generation process is inherently trajectory-based: the model iteratively transforms an initial noise sample into a final image. When the dataset discards the information that defines this trajectory, any DPO-style method must reconstruct or approximate the missing latent path in order to perform step or trajectory level optimization.

Prior diffusion-DPO methods commonly draw an independent noise sample and use a forward noising rule to generate intermediate latents, thereby estimating reverse-process quantities. But in diffusion, the true reverse trajectories are stochastic and typically *curved*, and sampling the exact reverse path conditional on an endpoint is not tractable; approximating it using forward noise injection can lead to a mismatch between the training surrogate and what the model actually does at inference. This mismatch can manifest as training instability, inefficient credit assignment, and a larger effective "decision space" for reward allocation.

Our key motivation is that rectified flow is structurally different and offers a simpler, more faithful estimator. *(i) RF trajectories are near-straight.* Rectified flow defines a coupling between data and prior that induces trajectories well-approximated by straight-line interpolation between endpoints. RF sampling is indexed by prior noise. v(ii) For a fixed prompt, different prior noises correspond to different trajectories and different final images. Thus, the prior noise is not incidental bookkeeping and it is a critical part of the trajectory identity. *(iii) Post-training is trajectory adaptation.* Pretraining constructs a general trajectory field; preference alignment should adapt this field so that, for typical prior noises, the induced trajectories yield human-preferred outcomes on a target data distribution.

These observations imply a simple but impactful change: store the prior noise together with generated image during dataset construction. If we have the endpoint pair that was actually used to sample the image, then the RF straightness property enables a cheap and faithful approximation of intermediate latents via interpolation. Based on this, we propose Prior Noise-Aware Preference Optimization (`PNAPO`), an off-policy alignment framework for rectified flow with two main contributions:

- Noise-augmented off-policy preference data. We build a preference dataset whose samples are sextuples, containing both winner and loser (prior noise, image) pairs, plus a continuous reward gap. This explicitly retains trajectory identity information absent in prior datasets.

- RF-consistent trajectory estimation and dynamic optimization. Using noise–image interpolation, `PNAPO` defines a DPO-style objective that compares policy and reference models on the same endpoint-conditioned intermediate states. We further introduce a dynamic regularization schedule that scales updates based on reward-gap difficulty and training stage, improving the training stability.

`PNAPO` is intentionally positioned as an **offline, RL-free** alternative: it avoids the engineering and compute overhead of on-policy online RL rollouts while exploiting RF geometry to obtain a lower-variance preference-optimization surrogate. We provide theoretical analysis showing why conditioning on stored prior noise yields a tighter bound/estimator for the RF setting, and empirical results on FLUX.1-dev

and SD3-M demonstrating consistent gains across multiple preference and alignment benchmarks with large compute savings compared to Diffusion-DPO.

## 2. Related Works

**Text-to-Image Generative Models.** T2I synthesis (Esser et al., 2024; Podell et al., 2023; Rombach et al., 2021) has evolved from GANs (Esser et al., 2021) to diffusion models (Ho et al., 2020; Song et al., 2020) recently, to flow-matching (Lipman et al., 2022) and rectified flow (Liu et al., 2022) formulations. RF models can be viewed as learning velocity fields along continuous-time trajectories between a Gaussian prior and the data distribution. Compared to standard diffusion, RF often yields more structured trajectories that are amenable to interpolation-based reasoning. Our work focuses on post-training alignment of such RF-based T2I models.

**Preference Optimization of Diffusion Models.** Supervised fine-tuning (SFT) dominates preference alignment in diffusion models. Inspired by RL-based LLM fine-tuning (Azar et al., 2024; Ethayarajh et al., 2024; Hong et al., 2024a; Schulman et al., 2017; Song et al., 2024), researchers train reward models (Kirstain et al., 2023; Wu et al., 2023) to mimic human judgment. DRaFT (Clark et al., 2023) and AlignProp (Prabhudesai et al., 2023) use differentiable rewards with backpropagation, while DPOK (Fan et al., 2023) and DDPO (Black et al., 2023) treat sampling as a MDP. Diffusion-DPO (Wallace et al., 2024) and D3PO (Yang et al., 2024a) optimize preferences at each denoising step, with variants like DenseReward (Yang et al., 2024b) focusing on early steps and Diffusion-KTO (Li et al., 2024) using binary feedback. SPO (Liang et al., 2025) aligns preferences throughout denoising process while InPO (Lu et al., 2025b) and SmPO (Lu et al., 2025c) employs DDIM Inversion (Mokady et al., 2023) and to optimize specific latent variables. In a related line of work, Diffusion-NPO (Wang et al., 2025) investigate the effectiveness of classifier-free guidance (CFG), training a model specifically calibrated to undesirable examples in order to steer sampling away from negative-conditional inputs. Although specialized variants (Croitoru et al., 2024; Dang et al., 2025; Hong et al., 2024b; Karthik et al., 2024; Lee et al., 2025b; Na et al., 2024; Lu et al., 2025d) exist, most approaches focus on conventional diffusion models. Current rectified flow methods typically just replace noise with velocity prediction (Liu et al., 2025b; Ma et al., 2025). While this demonstrates some effectiveness, it fails to account for the properties inherent to rectified flow, where the prior noise plays a critical role in post-training.

**Online Preference Alignment.** Recent methods adopt online RL or direct reward optimization (Xu et al., 2023) to continuously sample from the updated policy, e.g., GRPO-

family (Liu et al., 2025a; Xue et al., 2025; Li et al., 2025). These methods can achieve strong alignment but require substantial on-policy sampling and careful tuning to avoid instability. `PNAPO` targets a complementary regime: offline preference optimization where we can generate and store data once and then perform stable RL-free updates without continuous online rollouts. This design choice is particularly attractive when training compute, latency, or engineering constraints make online RL impractical.

## 3. Preliminaries

**Flow Matching and Diffusion Models.** Flow matching (Lipman et al., 2022) connects a data distribution $\boldsymbol{x}_0 \sim p_0$ and a noise distribution $\boldsymbol{x}_T \sim p_T$ ($\mathcal{N}(\mathbf{0}, \mathbf{I})$), learning a coupling $\pi(p_0, p_T)$ via an ODE $d\boldsymbol{x}_t = v(\boldsymbol{x}_t, t)dt$, on $t \in [0, T]$, where $v$ is parameterized by a network $v_\theta$. Contemporary methods define conditional paths $p_t(\boldsymbol{x}_t | \boldsymbol{x}_T)$ and fields $u_t(\boldsymbol{x}_t | \boldsymbol{x}_T)$, marginalizing over $p_0$ and $p_T$ to recover $p_t$ and $u_t$, with *Conditional Flow Matching* training objective:

$$\mathcal{L}_{\text{CFM}} = \mathbb{E}_{t, \boldsymbol{x}_t \sim p_t(\cdot | \boldsymbol{x}_T), \boldsymbol{x}_T \sim p_T} \| v_\theta(\boldsymbol{x}_t, t) - u_t(\boldsymbol{x}_t | \boldsymbol{x}_T) \|_2^2, \tag{1}$$

where $\boldsymbol{x}_t = a_t \boldsymbol{x}_0 + b_t \boldsymbol{x}_T$. We can express the optimization objective in the following format for *Diffusion Models*:

$$\mathcal{L}_{\text{Diffusion}} = \mathbb{E}_{t, \boldsymbol{x}_t \sim p_t(\cdot | \boldsymbol{x}_T), \boldsymbol{x}_T \sim p_T} w_t \lambda_t' \| \epsilon_\theta(\boldsymbol{x}_t, t) - \epsilon \|_2^2, \tag{2}$$

where $w_t = -\frac{1}{2} \lambda_t' b_t^2$ matches with $\mathcal{L}_{\text{CFM}}$ by applying $u_t(\boldsymbol{x}_t | \boldsymbol{x}_T) = \frac{a_t'}{a_t} \boldsymbol{x}_t - \frac{b_t}{2} \lambda_t' \boldsymbol{x}_T$, $\epsilon_\theta := \frac{2}{\lambda_t' b_t} (\frac{a_t'}{a_t} \boldsymbol{x}_t - v_\theta)$ and $\epsilon = \boldsymbol{x}_T$. *Rectified flow* establishes the forward trajectory as a straight-line path between data distribution and Gaussian:

$$\boldsymbol{x}_t = (1 - t)\boldsymbol{x}_0 + t\boldsymbol{x}_T, \tag{3}$$

and uses $\mathcal{L}_{\text{CFM}}$ which then corresponds to $w_t^{\text{RF}} = \frac{t}{1-t}$.

**DPO for Diffusion Models.** Preference datasets $\mathcal{D}(\boldsymbol{c}, \boldsymbol{x}_0^w, \boldsymbol{x}_0^l)$ contain human-ranked pairs: a prompt $\boldsymbol{c}$, winning image $\boldsymbol{x}_0^w$, and losing image $\boldsymbol{x}_0^l$. RLHF adapts the BT model (Bradley & Terry, 1952) via maximum likelihood estimation on $\mathcal{D}$. In diffusion models, recent work (Wallace et al., 2024) reformulates the optimization problem, resulting in a tractable surrogate:

$$\mathcal{L}_{\text{DPO}-\text{Diffusion}} := -\mathbb{E}_{(\boldsymbol{c}, \boldsymbol{x}_0^w, \boldsymbol{x}_0^l) \sim \mathcal{D}} \log \sigma$$
$$\left( \beta \mathbb{E}_{\substack{\boldsymbol{x}_{1:T}^w \sim p_\theta^{\boldsymbol{c}}(\boldsymbol{x}_{1:T}^w | \boldsymbol{x}_0^w) \\ \boldsymbol{x}_{1:T}^l \sim p_\theta^{\boldsymbol{c}}(\boldsymbol{x}_{1:T}^l | \boldsymbol{x}_0^l)}} \left[ \log \frac{p_\theta^{\boldsymbol{c}}(\boldsymbol{x}_{0:T}^w)}{p_{\text{ref}}^{\boldsymbol{c}}(\boldsymbol{x}_{0:T}^w)} - \log \frac{p_\theta^{\boldsymbol{c}}(\boldsymbol{x}_{0:T}^l)}{p_{\text{ref}}^{\boldsymbol{c}}(\boldsymbol{x}_{0:T}^l)} \right] \right) \tag{4}$$

For brevity, we denote $p_\theta(\cdot | \boldsymbol{c})$ as $p_\theta^{\boldsymbol{c}}(\cdot)$. Their estimation of Equ. 4 relies on the following expression:

$$\mathcal{L}(\theta) = -\mathbb{E}_{\mathcal{D}} \log \sigma(-\beta(\boldsymbol{s}_\theta^t(\boldsymbol{x}_0^w, \boldsymbol{c}) - \boldsymbol{s}_\theta^t(\boldsymbol{x}_0^l, \boldsymbol{c}))), \tag{5}$$

where $s_\theta^t(\boldsymbol{x}_0^*, \boldsymbol{c}) = \|\epsilon^* - \epsilon_\theta^t(\boldsymbol{x}_t^*, \boldsymbol{c})\|_2^2 - \|\epsilon^* - \epsilon_{\text{ref}}^t(\boldsymbol{x}_t^*, \boldsymbol{c})\|_2^2$ and $\epsilon^*$ is randomly sampled from $\mathcal{N}(\mathbf{0}, \mathbf{I})$ during training.

# 4. Method

In this section, we present the details of our PNAPO, an off-policy alignment approach for self-improving rectified flows. First, we introduce a novel fine-grained preference dataset collection method that incorporates prior noise. Then we provide a RF-consistent preference objective using noise–image interpolation and theoretical insights into its mechanism. Finally, we introduce a dynamic regularization schedule for stable and efficient training.

## 4.1. Off-Policy Data Construction

Given a reference policy model, our PNAPO first constructs fine-grained preference labels augmented with prior noise. The key insight is that post-training should focus on trajectory-specific refinement, where trajectories are shaped by prior noise. The off-policy dataset construction involves three steps: (1) Prompt Preparation, (2) Prior Noise-Image Pair Generation, and (3) Fine-Grained Label Collection.

**Step-1: Prompt Preparation.** We use DiffusionDB (Wang et al., 2022), a large-scale T2I dataset with 1.8 million real-world user prompts. Our sampling process involves: (1) *NSFW Filtering*: removing prompts with high Detoxify (Hanu & Unitary team, 2020) scores (retaining 83.67%). (2) *Deduplication*: applying text-based (Jaccard similarity $> 0.8$) and semantic (CLIP (Radford et al., 2021) cosine similarity $> 0.8$) deduplication. (3) *Cluster-based Resampling*: balancing semantic coverage by sampling proportionally from 100 KNN clusters. The final refined dataset contains 20k clean and diverse prompts.

**Step-2: Prior Noise-Image Pair Generation.** Using the prompt dataset from Step-1, we input the prompts into a T2I rectified flow base model. For each prompt, we sample a noise pair from a standard normal distribution and generate the corresponding image pair. Unlike traditional preference datasets that discard prior noise, we retain it as useful training information. Notably, we use the fine-tuned model itself as the base, ensuring stable preference alignment.

**Step-3: Fine-Grained Label Collection.** For training consistency, we use a pre-trained reward model HPSv2.1 to provide preference feedback. The score difference $\delta r$ between winner ($\boldsymbol{x}_0^w$) and loser ($\boldsymbol{x}_0^l$) images is computed as:

$$\delta r = r_\theta(\boldsymbol{x}_0^w) - r_\theta(\boldsymbol{x}_0^l), \qquad (6)$$

where $r_\theta(\boldsymbol{x}_0^*)$ is the reward model's scalar output. This approach pseudo-labels the dataset with interpretable and continuous feedback, acting as both a proxy for human preferences and a data cleanser. $\delta r$ captures nuanced perceptual distinctions (e.g., "slightly" vs. "significantly better"), guiding iterative updates more effectively.

## 4.2. RF-Consistent Optimization via Prior Noise

To optimize Equation 4, the key challenge lies in sampling $\boldsymbol{x}_{1:T} \sim p_\theta^{\boldsymbol{c}}(\boldsymbol{x}_{1:T}|\boldsymbol{x}_0)$ effectively; however, this sampling process is inherently intractable. To address this, we propose a reformulation of Equation 4 with prior noise $p_\theta(\boldsymbol{x}_T^*|\boldsymbol{x}_0^*)$:

$$\mathcal{L}(\theta) = -\mathbb{E}_\mathcal{D} \log \sigma \bigg( \beta \mathbb{E}_{\substack{\boldsymbol{x}_T^w \sim p_\theta(\boldsymbol{x}_T^w|\boldsymbol{x}_0^w) \\ \boldsymbol{x}_T^l \sim p_\theta(\boldsymbol{x}_T^l|\boldsymbol{x}_0^l)}} \mathbb{E}_{\substack{\boldsymbol{x}_{1:T-1}^w \sim p_\theta^{\boldsymbol{c}}(\cdot|\boldsymbol{x}_0^w, \boldsymbol{x}_T^w) \\ \boldsymbol{x}_{1:T-1}^l \sim p_\theta^{\boldsymbol{c}}(\cdot|\boldsymbol{x}_0^l, \boldsymbol{x}_T^l)}}$$
$$\left[ \log \frac{p_\theta^{\boldsymbol{c}}(\boldsymbol{x}_{0:T}^w)}{p_{\text{ref}}^{\boldsymbol{c}}(\boldsymbol{x}_{0:T}^w)} - \log \frac{p_\theta^{\boldsymbol{c}}(\boldsymbol{x}_{0:T}^l)}{p_{\text{ref}}^{\boldsymbol{c}}(\boldsymbol{x}_{0:T}^l)} \right] \bigg). \tag{7}$$

In contrast to Diffusion-DPO's approach of modeling $p_\theta(\boldsymbol{x}_T^*|\boldsymbol{x}_0^*)$ as the forward process $q(\boldsymbol{x}_T^*|\boldsymbol{x}_0^*) = q(\boldsymbol{x}_T^*)$, where $\boldsymbol{x}_T^*$ is drawn from an independent standard normal distribution $\mathcal{N}(\mathbf{0}, \mathbf{I})$ independent of $\boldsymbol{x}_0^*$, our $\boldsymbol{x}_T^w, \boldsymbol{x}_T^l$ are from the static dataset, which retains $p_\theta(\boldsymbol{x}_T^*|\boldsymbol{x}_0^*)$. Given $\boldsymbol{x}_T^*$, $p_\theta^{\boldsymbol{c}}(\boldsymbol{x}_{1:T-1}^*|\boldsymbol{x}_0^*, \boldsymbol{x}_T^*)$ becomes tractable if we estimate it using $p_\theta^{\boldsymbol{c}}(\boldsymbol{x}_{1:T-1}^*|\boldsymbol{x}_T^*)$, though this approach is evidently resource-intensive. Leveraging the straightness of rectified flow's sampling trajectories, we instead estimate $p_\theta^{\boldsymbol{c}}(\boldsymbol{x}_{1:T-1}^*|\boldsymbol{x}_0^*, \boldsymbol{x}_T^*)$ using an interpolation-based approximation $q(\boldsymbol{x}_{1:T-1}^*|\boldsymbol{x}_0^*, \boldsymbol{x}_T^*)$, yielding the following equation:

$$\mathcal{L}(\theta) = -\mathbb{E}_\mathcal{D} \log \sigma \bigg( \beta T \mathbb{E}_t \mathbb{E}_{\substack{\boldsymbol{x}_T^w \sim p_\theta(\boldsymbol{x}_T^w|\boldsymbol{x}_0^w) \\ \boldsymbol{x}_T^l \sim p_\theta(\boldsymbol{x}_T^l|\boldsymbol{x}_0^l)}} \mathbb{E}_{\substack{\boldsymbol{x}_t^w \sim q(\cdot|\boldsymbol{x}_0^w, \boldsymbol{x}_T^w) \\ \boldsymbol{x}_t^l \sim q(\cdot|\boldsymbol{x}_0^l, \boldsymbol{x}_T^l)}}$$
$$\mathbb{E}_{\substack{\boldsymbol{x}_{t-1}^w \sim q(\cdot|\boldsymbol{x}_0^w, \boldsymbol{x}_t^w) \\ \boldsymbol{x}_{t-1}^l \sim q(\cdot|\boldsymbol{x}_0^l, \boldsymbol{x}_t^l)}} \left[ \log \frac{p_\theta^{\boldsymbol{c}}(\boldsymbol{x}_{t-1}^w|\boldsymbol{x}_t^w)}{p_{\text{ref}}^{\boldsymbol{c}}(\boldsymbol{x}_{t-1}^w|\boldsymbol{x}_t^w)} - \log \frac{p_\theta^{\boldsymbol{c}}(\boldsymbol{x}_{t-1}^l|\boldsymbol{x}_t^l)}{p_{\text{ref}}^{\boldsymbol{c}}(\boldsymbol{x}_{t-1}^l|\boldsymbol{x}_t^l)} \right] \bigg). \tag{8}$$

According to Jensen's inequality, we can derive:

$$\mathcal{L}(\theta) \leq -\mathbb{E}_{\mathcal{D},t} \mathbb{E}_{\substack{\boldsymbol{x}_T^w \sim p_\theta(\boldsymbol{x}_T^w|\boldsymbol{x}_0^w) \\ \boldsymbol{x}_T^l \sim p_\theta(\boldsymbol{x}_T^l|\boldsymbol{x}_0^l)}} \mathbb{E}_{\substack{\boldsymbol{x}_t^w \sim q(\cdot|\boldsymbol{x}_0^w, \boldsymbol{x}_T^w) \\ \boldsymbol{x}_t^l \sim q(\cdot|\boldsymbol{x}_0^l, \boldsymbol{x}_T^l)}} \log \sigma \bigg( -\beta$$
$$\bigg( +\mathbb{D}_{\text{KL}}(q(\boldsymbol{x}_{t-1}^w|\boldsymbol{x}_0^w, \boldsymbol{x}_t^w)\|p_\theta^{\boldsymbol{c}}(\boldsymbol{x}_{t-1}^w|\boldsymbol{x}_t^w))$$
$$-\mathbb{D}_{\text{KL}}(q(\boldsymbol{x}_{t-1}^w|\boldsymbol{x}_0^w, \boldsymbol{x}_t^w)\|p_{\text{ref}}^{\boldsymbol{c}}(\boldsymbol{x}_{t-1}^w|\boldsymbol{x}_t^w))$$
$$-\mathbb{D}_{\text{KL}}(q(\boldsymbol{x}_{t-1}^l|\boldsymbol{x}_0^l, \boldsymbol{x}_t^l)\|p_\theta^{\boldsymbol{c}}(\boldsymbol{x}_{t-1}^l|\boldsymbol{x}_t^l))$$
$$+\mathbb{D}_{\text{KL}}(q(\boldsymbol{x}_{t-1}^l|\boldsymbol{x}_0^l, \boldsymbol{x}_t^l)\|p_{\text{ref}}^{\boldsymbol{c}}(\boldsymbol{x}_{t-1}^l|\boldsymbol{x}_t^l)) \bigg) \bigg) \tag{9}$$

Through parameterization of the rectified flow reverse process, the aforementioned loss simplifies to:

$$\mathcal{L}_{\text{PNAPO}}(\theta) = -\mathbb{E}_{(\boldsymbol{c}, \boldsymbol{x}_0^w, \boldsymbol{x}_0^l, \boldsymbol{x}_T^w, \boldsymbol{x}_T^l) \sim \mathcal{D}_{\text{PNAPO}}, t}$$
$$\log \sigma(-\beta(s_\theta^t(\boldsymbol{x}_0^w, \boldsymbol{x}_T^w, \boldsymbol{c}) - s_\theta^t(\boldsymbol{x}_0^l, \boldsymbol{x}_T^l, \boldsymbol{c}))) \tag{10}$$

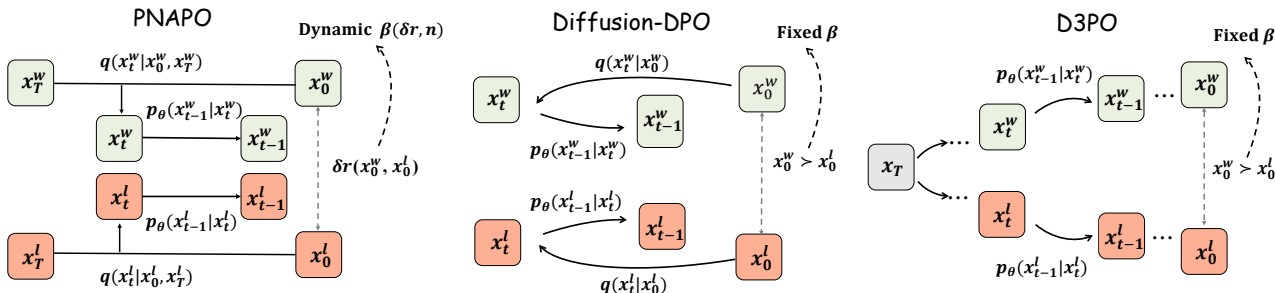

*Figure 3.* **Comparison of `PNAPO` versus DPO baselines.** Compared to Diffusion-DPO's stochastic noise injection, PNAPO employs a prior noise with $x_T$-$x_0$ interpolation for more accurate estimation, while surpassing D3PO in efficiency by avoiding iterative reverse processes. Additionally, dynamic regularization generation leverages $\delta r$ reward gaps and training step $n$.

where $t \sim \mathcal{U}(0, T)$ and we define the $s_\theta^t$ as:

$$
\begin{aligned}
s_\theta^t(x_0^*, x_T^*, c) = &\|(x_T^* - x_0^*) - v_\theta(x_t^*, t, c)\|_2^2 \\
&- \|(x_T^* - x_0^*) - v_{\text{ref}}(x_t^*, t, c)\|_2^2,
\end{aligned} \quad (11)
$$

where $x_t^* = (1 - t)x_0^* + tx_T^*$. Similar to the delayed feedback/sparse reward problem in RL, Diffusion-DPO faces analogous challenges for its forward noise-addition strategy. Our method significantly reduces the decision space, substantially improving training efficiency.

**Why `PNAPO` is better than Diffusion-DPO?** Notably, while Diffusion-DPO employs the forward process $q(x_{1:T}|x_0)$ to estimate the reverse process $p_\theta(x_{1:T}|x_0)$, our method utilizes $p_\theta(x_T|x_0)q(x_{1:T-1}|x_0, x_T)$ for estimation. This approximation yields lower error since

$$
\begin{aligned}
&\mathbb{D}_{\text{KL}}(p_\theta(x_T|x_0)q(x_{1:T-1}|x_0, x_T)||p_\theta(x_{1:T}|x_0)) \\
=&\mathbb{D}_{\text{KL}}(q(x_{1:T-1}|x_0, x_T)||p_\theta(x_{1:T-1}|x_0, x_T)) \quad (12) \\
\leq&\mathbb{D}_{\text{KL}}(q(x_{1:T}|x_0)||p_\theta(x_{1:T}|x_0)).
\end{aligned}
$$

### 4.3. Dynamic Regularization

Current preference alignment approaches for diffusion models largely overlook the dynamics during fine-tuning. Specifically, conventional DPO suffers from two key limitations: (1) it uniformly treats all image pairs, ignoring variations in their learning difficulty (e.g., subtle vs. obvious quality gaps), which leads to improper gradient scaling. (2) The fixed regularization term increasingly impedes model updates as training progresses, and accordingly PNAPO introduces a dynamic training strategy. To gain mechanistic insight into alignment dynamics, analyzing the loss function's gradient proves particularly instructive. The gradient with respect to parameters $\theta$ can be decomposed as follows:

$$
\begin{aligned}
\nabla_\theta \mathcal{L}_{\text{PNAPO}}(\theta) = \mathbb{E}_{(c, x_0^w, x_0^l, x_T^w, x_T^l) \sim \mathcal{D}_{\text{PNAPO}}, t} \\
\left[ \beta\sigma\big( -\beta s_\theta^t(x_0^w, x_T^w, c) + \beta s_\theta^t(x_0^l, x_T^l, c)) \right) \\
\left[ \nabla_\theta s_\theta^t(x_0^w, x_T^w, c) - \nabla_\theta s_\theta^t(x_0^l, x_T^l, c) \right] \Big].
\end{aligned} \quad (13)
$$

Intuitively, the loss increases the likelihood of generating winning images while decreasing losing ones. Crucially, gradient scale depends on: (1) the regularization coefficient $\beta$, and (2) the margin (the $\sigma(\cdot)$ value). *Fixed $\beta$ fails to adapt to varying image pair importance.* Conversely, when the margin is negative, increasing $\beta$ enlarges the margin, which accelerates the model's alignment with winner images while promoting divergence from the reference model. However, with positive margins (indicating good training), increasing $\beta$ conversely reduces the margin, yielding smaller updates. *As training progresses, strong regularization gradually pulls the model back toward the reference model.* This motivates our dynamic regularization $\beta(\delta r, n)$:

$$
\beta(\delta r, n) = \beta \cdot f(\delta r) \cdot g(n). \quad (14)
$$

Here training sample controller $f$ must increase monotonically to 1, where $\delta r \in [0, +\infty)$ and training process controller $g$ decays as a annealing factor. These are defined as:

$$
f(\delta r) = 2 \cdot \sigma(\delta r) - 1
$$

$$
g(n) = \begin{cases} 1, & \text{if } n \leq n_1, \\ \frac{1}{2} + \frac{1}{2} \cdot \cos(\frac{1}{2} \cdot \frac{n-n_1}{n_2-n_1}\pi), & \text{if } n_1 < n < n_2, \\ \frac{1}{2}, & \text{if } n \geq n_2. \end{cases}
\quad (15)
$$

Here $\sigma$ denotes the sigmoid function, $n$ represents the training step, and $n_1, n_2$ are user-defined thresholds. The function $f(\delta r)$ links $\beta(\delta r, n)$ to reward difference $\delta r$: when the margin is negative, increasing $\delta r$ raises $\beta(\delta r, n)$ to accelerate training; otherwise, the opposite effect occurs. Meanwhile, $g(n)$ starts high in early training, then gradually decreases for $n > n_1$, halving by $n = n_2$.

## 5. Experiments

### 5.1. Experimental Setup

**Implementation Details.** We employ FLUX.1-dev (FLUX) and Stable Diffusion 3 Medium (SD3-M) as our rectified

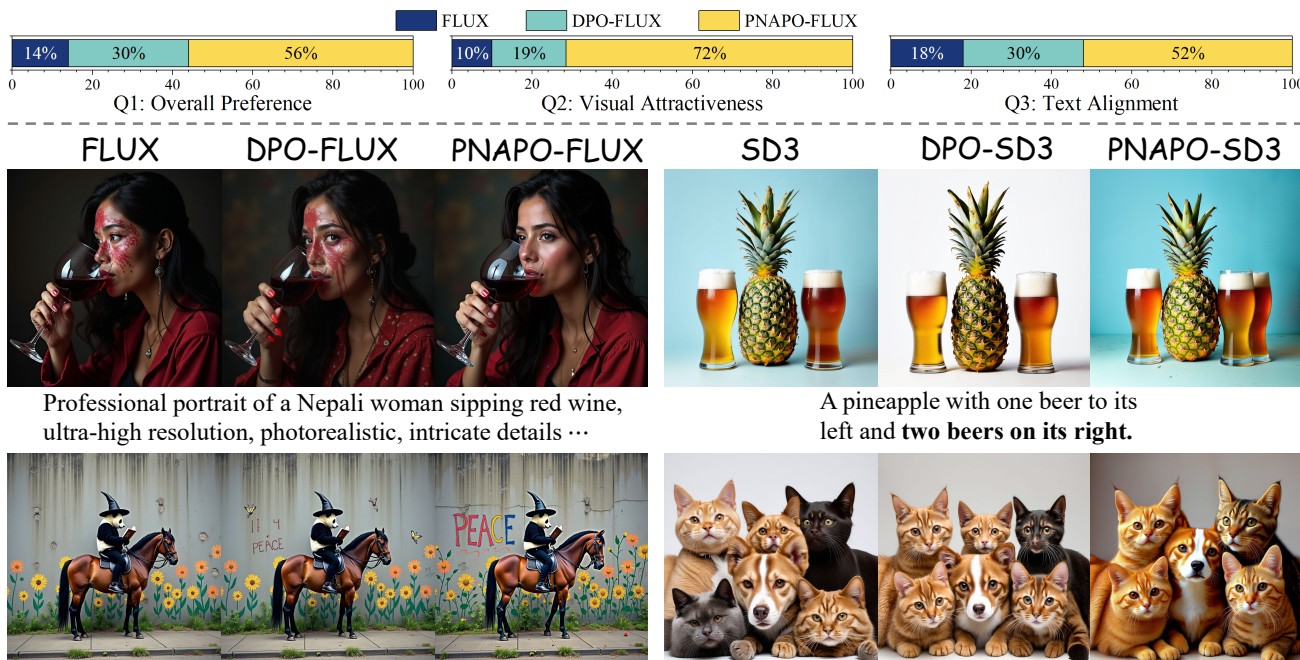

*Figure 4.* **User Study and Qualitative Comparison.** Top, human evaluations show PNAPO-FLUX significantly outperforming DPO-FLUX and the base FLUX model. Bottom, we present qualitative comparisons between PNAPO and Diffusion-DPO when applied to the FLUX and SD3-M. The results demonstrate that our model achieves superior image generation quality.

flow models for T2I generation. For each model, we utilize 20,000 prompts from DiffusionDB, generating two images per prompt. Image generation with both FLUX and SD3-M is performed using the Euler discrete scheduler with a guidance scale of 1 over 50 sampling steps. To ensure fair comparison of training efficiency, all baselines employ identical hyperparameters. We adopt AdamW as the optimizer for both FLUX and SD3-M with a learning rate of $1e^{-6}$. All experiments are conducted on 8 NVIDIA H800 GPUs. For FLUX training, $\beta$ is set to 2000, while for SD3-M, it is set to 5000. All experimental details are comprehensively documented in the Appendix.

**Evaluation.** We evaluate the model using multiple metrics: PickScore (Kirstain et al., 2023), HPSv2.1 (Wu et al., 2023), LAION aesthetic classifier and ImageReward (Xu et al., 2023) for simulating human preference; CLIP (Radford et al., 2021) for measuring text alignment; and T2I benchmark GenEval (Ghosh et al., 2023) for object-focused generation. We compare the following baselines: Diffusion-DPO (Wallace et al., 2024), Supervised Fine-Tuning (SFT), IPO (Azar et al., 2024), and CaPO (Lee et al., 2025b). To guarantee an unbiased evaluation, we faithfully reproduce Diffusion-DPO, SFT, and IPO with identical hyperparameters and model configurations. During evaluation, we employ the HPDv2 (Wu et al., 2023) and OPDv1 (is Better-Together, 2025) as test sets, using the median reward score

*Table 1.* **Computational cost comparison.** We report the NVIDIA H800 GPU hours required for training our PNAPO and the DPO-Diffusion on SD3-M and FLUX.

| Model | GPU-Hours | Model | GPU-Hours |
|---|---|---|---|
| DPO-SD3 | ∼ 249.6 | DPO-FLUX | ∼ 422.4 |
| PNAPO-SD3 | ∼ **20.8** | PNAPO-FLUX | ∼ **35.2** |

and win rate as preference metrics.

**5.2. Primary Results**

**Qualitative Results.** As demonstrated in Figures 2 and 4, the proposed PNAPO consistently outperforms existing baseline approaches across multiple dimensions, including text-image alignment, visual aesthetics, and photorealism. In particular, PNAPO effectively mitigates characteristic artifacts such as background blurring often observed in FLUX-generated samples, as clearly illustrated in Figure 2. When compared against competitive methods such as Diffusion-DPO, our approach yields higher-quality outputs on both SD3-M and FLUX architectures, with noticeable improvements in textual fidelity and overall visual appeal. These qualitative enhancements align closely with human preferences, reinforcing the practical advantages of PNAPO.

**User Study.** We conduct a user study involving 10 partici-

*Table 2.* **Quantitive Comparison.** We utilize the HPDv2 and OPDv1 prompt datasets to generate images with both the SD3-M and FLUX. Each reward model is evaluated, and we present the median reward score (Score) alongside the win-rate (WR) of our PNAPO against baselines. Superior performance is indicated by higher scores and win-rates. In the Score column, the top value is **bold**. Win-rates surpassing 50% are underlined. We replicate the baselines under the exact same experimental configuration.

| Model | HPDv2 (3200 prompts) | | | | | | | | | | OPDv1 (7459 prompts) | | | | | | | | | |
| | PickScore↑ | | HPSv2.1↑ | | ImReward↑ | | Aesthetic↑ | | CLIP↑ | | PickScore↑ | | HPSv2.1↑ | | ImReward↑ | | Aesthetic↑ | | CLIP↑ | |
| | Score | WR | Score | WR | Score | WR | Score | WR | Score | WR | Score | WR | Score | WR | Score | WR | Score | WR | Score | WR |
| SD3-M | 22.68 | 66.6 | 30.75 | 70.8 | 1.306 | 60.4 | 5.949 | 64.2 | 33.01 | 61.9 | 22.06 | 72.3 | 31.96 | 78.9 | 1.383 | 60.8 | 6.287 | 70.8 | 34.73 | 66.4 |
| SFT | 22.76 | 61.5 | 30.83 | 70.0 | 1.367 | 54.8 | 5.978 | 60.0 | 33.17 | 60.1 | 22.18 | 59.2 | 32.10 | 74.7 | 1.435 | 53.3 | 6.312 | 66.1 | 34.87 | 63.5 |
| DPO | 22.74 | 63.6 | 31.13 | 67.5 | 1.353 | 55.7 | 5.988 | 59.3 | 33.24 | 58.7 | 22.13 | 68.8 | 32.39 | 70.4 | 1.405 | 56.4 | 6.295 | 69.1 | 34.99 | 60.8 |
| IPO | 22.73 | 65.3 | 30.92 | 69.7 | 1.364 | 55.1 | 5.976 | 61.1 | 33.26 | 58.6 | 22.19 | 59.1 | 32.33 | 70.8 | 1.411 | 55.0 | 6.313 | 61.4 | 35.02 | 60.6 |
| PNAPO | **22.85** | - | **31.62** | - | **1.387** | - | **6.069** | - | **33.65** | - | **22.37** | - | **33.09** | - | **1.465** | - | **6.414** | - | **35.58** | - |
| FLUX | 22.95 | 67.4 | 30.50 | 79.0 | 1.175 | 57.0 | 6.299 | 75.8 | 34.05 | 60.2 | 22.17 | 77.5 | 30.74 | 84.7 | 1.202 | 58.8 | 6.550 | 73.3 | 35.97 | 68.2 |
| SFT | 23.09 | 56.7 | 29.99 | 88.4 | 1.115 | 63.6 | 6.358 | 69.9 | 34.23 | 59.4 | 22.32 | 61.5 | 30.06 | 90.9 | 1.135 | 68.0 | 6.585 | 67.5 | 36.16 | 65.8 |
| DPO | 22.97 | 66.1 | 30.84 | 78.6 | 1.185 | 56.4 | 6.307 | 75.6 | 34.64 | 55.7 | 22.20 | 76.6 | 30.79 | 84.6 | 1.209 | 57.6 | 6.548 | 73.7 | 36.19 | 65.1 |
| IPO | 22.98 | 65.3 | 30.87 | 78.1 | 1.174 | 55.3 | 6.311 | 75.0 | 34.60 | 56.0 | 22.24 | 73.8 | 30.91 | 81.1 | 1.212 | 56.3 | 6.574 | 70.2 | 36.22 | 64.7 |
| PNAPO | **23.19** | - | **31.71** | - | **1.217** | - | **6.475** | - | **34.71** | - | **22.52** | - | **32.10** | - | **1.238** | - | **6.692** | - | **36.89** | - |

pants, with results summarized in Figure 4. Each participant evaluated 20 randomly selected image pairs, comparing PNAPO-FLUX against several strong baselines. The evaluation focused on three key criteria: (1) overall preference, (2) visual appeal, and (3) text-image alignment. Our method achieved superior results across all categories, attaining 56% in overall preference, 72% in visual appeal, and 52% in text alignment. These outcomes statistically affirm the effectiveness of PNAPO and its alignment with human judgment in real-world visual quality assessment.

**Quantitative Results on Text-Image Alignment.** For text-image alignment evaluation, we benchmark on *GenEval*, a specialized object-generation dataset, comparing against: (1) base models (SD3-M, FLUX) and (2) SOTA preference-aligned baselines (DPO-aligned variants and CaPO-aligned SD3-M). Table 3 shows PNAPO consistently improves alignment metrics, boosting SD3-M from 0.68 to 0.73 (+7.4%) and FLUX from 0.65 to 0.69 (+6.2%). This represents a 2.8% and 4.5% absolute improvement over CaPO-SD3-M (0.71) and DPO-FLUX (0.66) respectively, demonstrating both higher performance and better cross-architectural generalization with our PNAPO.

**Quantitative Results on Preference Alignment.** Table 2 presents the preference reward scores of our PNAPO models against baseline models, along with their comparative win rates. Overall, our PNAPO fine-tuned SD3-M and FLUX models demonstrate superior performance across all test datasets and reward scores compared to the baselines. Notably, on the OPDv1 test set, PNAPO-SD3-M and PNAPO-FLUX achieve median HPSv2.1 reward scores of 33.09 and 32.10, surpassing their original counterparts (SD3-M and FLUX) by +1.13 and +1.36, respectively. Furthermore, the HPSv2.1 preference metric reveal that PNAPO-FLUX achieves win rates of 84.6% against DPO-FLUX and 81.1%

*Table 3.* **GenEval Evaluation.** We evaluate PNAPO-SD3-M and PNAPO-FLUX on the T2I benchmark GenEval. Under PNAPO, both SD3-M and FLUX exhibit improved evaluation metrics. The top value is **bold** in each column.

| Model | GenEval | | | | | | |
| | Single | Two | Count | Attri. | Pos. | Color | OverAll |
| SD1.5 | 0.96 | 0.38 | 0.35 | 0.04 | 0.03 | 0.76 | 0.42 |
| SDXL | 0.97 | 0.70 | 0.41 | 0.22 | 0.10 | 0.87 | 0.55 |
| SD3.5-L | 0.99 | 0.88 | 0.62 | 0.52 | 0.25 | 0.82 | 0.68 |
| FLUX-S. | 0.98 | 0.80 | 0.57 | 0.35 | 0.24 | 0.63 | 0.60 |
| SD3-M | 0.99 | 0.84 | 0.56 | 0.52 | **0.32** | 0.84 | 0.68 |
| DPO | 0.99 | 0.85 | 0.60 | 0.56 | **0.32** | 0.84 | 0.69 |
| CaPO | 0.99 | **0.87** | 0.63 | 0.59 | 0.31 | **0.86** | 0.71 |
| PNAPO | **1.00** | **0.87** | **0.71** | **0.62** | **0.32** | **0.86** | **0.73** |
| FLUX | 0.98 | 0.77 | 0.72 | 0.42 | 0.20 | 0.78 | 0.65 |
| DPO | **0.99** | 0.79 | 0.73 | 0.44 | 0.22 | 0.78 | 0.66 |
| PNAPO | **0.99** | **0.84** | **0.76** | **0.48** | **0.24** | **0.81** | **0.69** |

against IPO-FLUX. Similar improvements are observed across other metrics, validating the effectiveness of PNAPO.

**Computational Cost.** During training, we use LoRA (Hu et al., 2022) for FLUX and full-parameter fine-tuning for SD3-M. Our PNAPO requires only 35.2 and 20.8 GPU (H800) hours for FLUX and SD3-M, respectively. Compared to Diffusion-DPO's 422.4 and 249.6 GPU hours, our PNAPO achieves 12× less training cost than Diffusion-DPO while significantly improving generation quality.

### 5.3. Ablation Studies and Analysis

**Proposed Improvements.** One cornerstone of our PNAPO involves sampling conditional noise from the target image

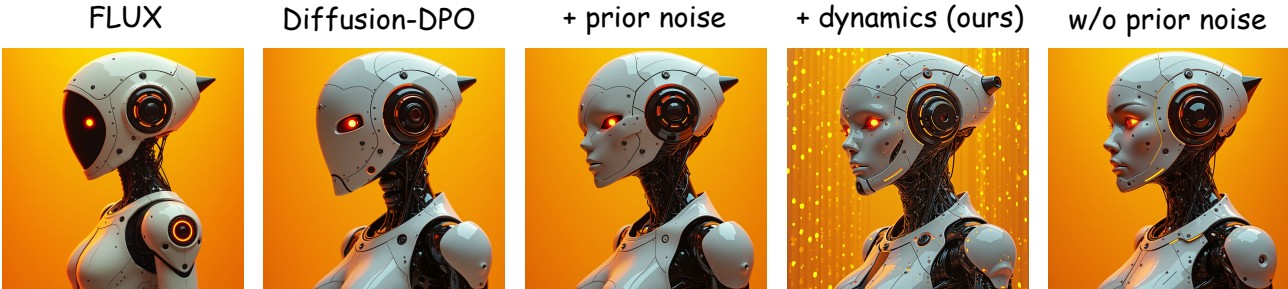

FLUX    Diffusion-DPO    + prior noise    + dynamics (ours)    w/o prior noise

A sleek, **futuristic robot** with glowing red eyes and accents, set against a vibrant orange and yellow background.

*Figure 5.* Qualitative ablation comparison of our proposed improvements.

*Table 4.* Ablation study for our improvements.

| | PickScore↑ | HPS↑ | ImReward↑ | Aesth.↑ | CLIP↑ |
|---|---|---|---|---|---|
| DPO | 22.97 | 30.84 | 1.185 | 6.307 | 34.64 |
| $+p_\theta(\boldsymbol{x}_T|\boldsymbol{x}_0)$ | 23.06 | 31.08 | 1.201 | 6.394 | 34.66 |
| +Dynamics | **23.19** | **31.71** | **1.217** | **6.475** | **34.71** |
| $-p_\theta(\boldsymbol{x}_T|\boldsymbol{x}_0)$ | 23.00 | 30.96 | 1.197 | 6.368 | 34.68 |

*Table 5.* Ablation study on regularization.

| KL Div. | PickScore↑ | HPS↑ | ImReward↑ | Aesth.↑ | CLIP↑ |
|---|---|---|---|---|---|
| Fixed $\beta$ | 23.06 | 31.08 | 1.201 | 6.394 | 34.68 |
| $\beta \cdot f(\delta r)$ | 23.16 | 31.66 | 1.212 | 6.461 | 34.68 |
| $\beta \cdot g(n)$ | 23.09 | 31.13 | 1.205 | 6.429 | 34.70 |
| $\beta(\delta r, n)$ | **23.19** | **31.71** | **1.217** | **6.475** | **34.71** |

*Table 6.* Ablation study on reward models.

| Reward | PickScore↑ | HPS↑ | ImReward↑ | Aesth.↑ | CLIP↑ |
|---|---|---|---|---|---|
| PickScore | 23.18 | 31.49 | 1.213 | 6.470 | 34.66 |
| HPSv2.1 | **23.19** | **31.71** | **1.217** | 6.475 | **34.71** |
| ImReward | 23.05 | 31.10 | 1.206 | 6.392 | 34.60 |
| Aesthetic | 23.10 | 31.23 | 1.204 | **6.509** | 34.57 |
| CLIP | 23.04 | 31.12 | 1.201 | 6.375 | 34.61 |

*Table 7.* Ablation study on hyperparameters.

| $(n_1, n_2)$ | PickScore↑ | HPS↑ | ImReward↑ | Aesth.↑ | CLIP↑ |
|---|---|---|---|---|---|
| (500,2000) | 23.01 | 30.96 | 1.198 | 6.355 | 34.64 |
| (1000,1500) | 23.17 | 31.68 | 1.212 | 6.465 | 34.66 |
| (1000,2000) | **23.19** | **31.71** | **1.217** | **6.475** | **34.71** |
| (1000,3000) | 23.17 | 31.67 | 1.211 | 6.459 | 34.70 |
| (1000,4000) | 23.16 | 31.62 | 1.210 | 6.456 | 34.69 |

in the dataset $p_\theta(\boldsymbol{x}_T|\boldsymbol{x}_0)$ and estimating the latent variable $\boldsymbol{x}_t$ through interpolation. Furthermore, we introduce dynamic regularization term $\beta(\delta r, n)$ to control the gradients of the loss function. As demonstrated in Figure 5 and Table 4, incorporating prior noise significantly enhances image generation quality. The implementation of dynamic training substantially improves model performance. Notably, even without prior noise, it delivers marked improvements over the DPO method. These results validate the individual effectiveness of both components in our approach. Through ablation studies on the regularization terms (Table 5), we observe both the training sample controller $f(\delta r)$ and process controller $g(n)$ independently contribute to performance enhancement, while their combination yields optimal results.

**Reward Model Selection.** As shown in Table 6, leveraging text-aware preference reward models (e.g., PickScore and HPS v2.1) for training guidance enhances both visual appeal and text-rendering fidelity. However, alternative reward models tend to prioritize optimizing specific metrics, such as aesthetic classifier, often at the expense of text fidelity. Our analysis reveals that reward models effectively function as pseudo-labeling mechanisms for dataset refinement. Notably, HPSv2.1, as an advanced model, demonstrates

superior performance across comprehensive metrics.

**Choices of Parameters.** Table 7 presents the impact of the parameters training steps threshold $(n_1, n_2)$ on performance. Our analysis reveals that reducing the regularization term degrades model effectiveness, while maintaining an strong regularization term gradually pulls the model back toward the reference model as training progresses. In our experiments, the configuration with $(n_1, n_2) = (1000, 2000)$ demonstrate optimal performance.

# 6. Conclusion

We introduced `PNAPO`, an offline, RL-free preference alignment method for rectified flow T2I models. `PNAPO` addresses the fact that standard preference datasets store only final image pairs and omit trajectory identity where each sample is tied to a specific prior noise. `PNAPO` enables endpoint-conditioned trajectory estimation via noise–image interpolation and yielding a lower-variance DPO-style objective than independent noising. We also use a dynamic regularization that scales updates by reward gap and training progress for improved stability and efficiency. Across FLUX and SD3-M and multiple benchmarks, `PNAPO` im-

proves alignment and fidelity while reducing training compute. Theoretical results are RF-specific, attributing gains to endpoint conditioning and RF straightness.

## Acknowledgments

This work was supported by the National Major Science and Technology Projects (the grant number 2022ZD0117000) and the National Natural Science Foundation of China (grant number 62202426). We thank Shanghai Institute for Mathematics and Interdisciplinary Sciences (SIMIS) for their financial support. This research was funded by SIMIS under grant number [SIMIS-ID-2025-AD]. The authors are grateful for the resources and facilities provided by SIMIS, which were essential for the completion of this work.

## Impact Statement

`PNAPO` is an offline, RL-free preference optimization method for rectified-flow text-to-image models that improves alignment and training efficiency by leveraging stored prior noise. Positively, it can reduce compute and engineering costs for post-training, making preference-based improvement more accessible and enabling faster iteration on quality and safety-related tuning. However, better-aligned and higher-quality generation can also increase misuse risks, including producing deceptive imagery (impersonation, propaganda), enabling privacy or copyright violations, and amplifying biases if preference signals or reward models encode skewed values. Because `PNAPO` relies on offline preference data, dataset and reward-model choices can systematically steer outputs toward biased stereotypes or "reward-hacked" artifacts. Mitigations include careful prompt/data curation, bias audits of reward models and labels, evaluation with multiple independent metrics and human review, and deployment safeguards such as content filtering and provenance/watermarking. `PNAPO` does not create fundamentally new capabilities, but it can lower the barrier to optimizing existing models, so responsible data and deployment practices remain essential.

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

# A. Background

## A.1. More Related Works

**Conditional Generative Models.** Diffusion models belong to a family of generative approaches that create data through an iterative denoising process. These models learn to reverse a predefined forward process that gradually adds noise to data. By capitalizing on neural networks' powerful function approximation capabilities, they can generate diverse samples that accurately reflect the training data distribution. The field primarily recognizes two fundamental formulations: denoising diffusion probabilistic models and score-based generative models, which provide complementary mathematical frameworks for the generation task. Recent advances in diffusion models, particularly innovations such as Rectified Flow, have established them as the prevailing methodology in generative modeling. These models demonstrate superior performance in both output quality and training stability when compared to previous generation techniques. Their success has led to significant breakthroughs across multiple domains including conditional image synthesis, audio generation, and video production. This study specifically examines their application in conditional image generation. Text prompts commonly serve as the primary guidance mechanism in such generation systems. Typically, a pretrained text encoder transforms linguistic inputs into embedding representations, enabling effective text-to-image translation. Our task involves improving model performance by leveraging the model's own text-image pair outputs, with preference optimization serving as the post-training enhancement strategy.

**Preference Optimization of Large Language Models.** Reinforcement Learning from Human Feedback (RLHF) has emerged as a fundamental paradigm for adapting large language models (LLMs) to human-aligned behaviors. The standard implementation follows a two-stage procedure: initial development of a preference model that captures human evaluation patterns, followed by policy optimization through reinforcement learning to maximize the predicted rewards. This approach has been successfully deployed in state-of-the-art systems like ChatGPT. The conventional RLHF pipeline employs Proximal Policy Optimization (PPO) (Schulman et al., 2017) as its core algorithm, which necessitates simultaneous operation of multiple model components - including the active policy, reference model, value function estimator, and reward predictor. However, PPO's computational intensity and complex optimization landscape frequently pose implementation challenges. To mitigate these issues, researchers have developed more efficient alternatives. Some approaches employ REINFORCE-derived algorithms within the RLHF framework, while others bypass conventional reinforcement learning altogether by leveraging reward-guided sample ranking for supervised fine-tuning. Notable examples include RAFT (reward-weighted supervised learning) (Dong et al., 2023), RRHF (ranking-based alignment) (Yuan et al., 2023), and rejection sampling techniques that select outputs from high-probability policy regions.

Recent innovations have further streamlined the alignment process. Direct Preference Optimization (DPO) (Rafailov et al., 2023) circumvents explicit reward modeling by directly optimizing the policy using implicit reward signals derived from the Bradley-Terry framework. The Identity Preference Optimization (IPO) (Azar et al., 2024) method challenges conventional approaches by demonstrating that pointwise reward maximization cannot fully capture pairwise preference structures, proposing instead a probability-based optimization scheme. ORPO introduces additional efficiency by unifying supervised fine-tuning and preference optimization without requiring a reference model. Alternative reward formulations have also expanded the methodological landscape. Kahneman-Tversky Optimization (KTO) (Ethayarajh et al., 2024) replaces preference likelihood maximization with prospect theory-inspired utility modeling, while Preference Ranking Optimization (PRO) (Song et al., 2024) enhances LLM training through comparative reward information. These advancements collectively represent significant progress in developing more efficient and theoretically grounded alignment techniques.

**Further Preference Optimization of Diffusion Models.** The application of preference alignment techniques extends well beyond text-to-image diffusion models (Eyring et al., 2024; Miao et al., 2024; Yuan et al., 2024; Liu et al., 2025c; Peng et al.; Dunlop et al., 2025; Simon et al., 2025; Guo et al., 2025; Jain et al., 2025; Na et al., 2025; Lu et al., 2025d), with various generative domains developing specialized approaches tailored to their distinct data structures. While these developments represent significant progress, human preference alignment in diffusion models (Hu et al., 2025a; Ren et al., 2025; Lyu et al., 2025; Wu et al., 2025; Fu et al., 2025; Hu et al., 2025b; Zhang et al., 2025) remains a nascent research area. Promising future directions (Cao et al., 2025a;b; Lin et al., 2026; 2025; Lu et al., 2025a; Wang et al., 2026) may involve transferring alignment methodologies from large language models to generative visual systems (Borso et al., 2025; Lee et al., 2025a; Zheng et al., 2025; Lu et al., 2025d), as well as expanding these techniques to novel sensory modalities including auditory and haptic domains.

# B. More Preliminaries

**Flow Matching and Diffusion Models.**   To For the construction of $u_t$, we define a forward process that characterizes a probability path $p_t$ between the initial distribution $p_0$ and the terminal normal distribution $p_1 = \mathcal{N}(\mathbf{0}, \mathbf{I})$ as:

$$\boldsymbol{x}_t = a_t \boldsymbol{x}_0 + b_t \boldsymbol{x}_T, \tag{16}$$

where $\boldsymbol{x}_T \sim \mathcal{N}(\mathbf{0}, \mathbf{I})$.

For $a_0 = 1$, $b_0 = 0$, $a_1 = 0$ and $b_1 = 1$, the marginals,

$$p_t(\boldsymbol{x}_t) = \mathbb{E}_{\boldsymbol{x}_T \sim \mathcal{N}(\mathbf{0}, \mathbf{I})} p_t(\boldsymbol{x}_t | \boldsymbol{x}_T), \tag{17}$$

are in accordance with the underlying data distribution and the prescribed noise distribution. To represent the relationship between $\boldsymbol{x}_t$, $\boldsymbol{x}_0$ and $\boldsymbol{x}_T$, we introduce $\phi_t$ and $u_t$ as:

$$\phi_t(\cdot | \boldsymbol{x}_T) : \boldsymbol{x}_0 \longrightarrow a_t \boldsymbol{x}_0 + b_t \boldsymbol{x}_T \tag{18}$$

$$u_t(\boldsymbol{x}_t | \boldsymbol{x}_T) = \phi'_t(\phi_t^{-1}(\boldsymbol{x}_t | \boldsymbol{x}_T) | \boldsymbol{x}_T) \tag{19}$$

Since $\boldsymbol{x}_t$ can be expressed as the solution to the ODE $\boldsymbol{x}'_t = u_t(\boldsymbol{x}_t | \boldsymbol{x}_T)$ with initial condition $\boldsymbol{x}_0$, where $u_t(\cdot | \boldsymbol{x}_T)$ generates the conditional probability path $p_t(\cdot | \boldsymbol{x}_T)$, we highlight that it is possible to construct a marginal vector field $u_t$ that induces the marginal probability path $p_t$, with he conditional vector fields $u_t(\cdot | \boldsymbol{x}_T)$:

$$u_t(\boldsymbol{x}_t) = \mathbb{E}_{\boldsymbol{x}_T \sim \mathcal{N}(\mathbf{0}, \mathbf{I})} u_t(\boldsymbol{x}_t | \boldsymbol{x}_T) \frac{p_t(\boldsymbol{x}_t | \boldsymbol{x}_T)}{p_t(\boldsymbol{x}_t)}. \tag{20}$$

While performing regression on $u_t$ through the *Flow Matching* objective function:

$$\mathcal{L}_{\text{FM}} = \mathbb{E}_{t, \boldsymbol{x}_t \sim p_t(\boldsymbol{x}_t),} \| v_\theta(\boldsymbol{x}_t, t) - u_t(\boldsymbol{x}_t) \|_2^2. \tag{21}$$

The marginalization inherent in these equations makes direct optimization intractable. We circumvent this difficulty by utilizing the conditional vector field $u_t(\boldsymbol{x}_t | \boldsymbol{x}_T)$, which offers an equivalent formulation as Equation 1 that is computationally manageable.

In order to derive an explicit expression for the loss, we perform the substitution $\phi'_t(\boldsymbol{x}_0 | \boldsymbol{x}_T) = a'_t \boldsymbol{x}_0 + b'_t \boldsymbol{x}_T$ and $\phi_t^{-1}(\boldsymbol{x}_t | \boldsymbol{x}_T) = \frac{\boldsymbol{x}_t - b_t \boldsymbol{x}_T}{a_t}$ into Equation 19:

$$\boldsymbol{x}'_t = u_t(\boldsymbol{x}_t | \boldsymbol{x}_T) = \frac{a'_t}{a_t} \boldsymbol{x}_t - b_t \left( \frac{a'_t}{a_t} - \frac{b'_t}{b_t} \right) \boldsymbol{x}_T. \tag{22}$$

Here we consider the signal-to-noise ratio $\lambda_t := \log \frac{a_t^2}{b_t^2}$. Thus we have $\lambda'_t = 2\left( \frac{a'_t}{a_t} - \frac{b'_t}{b_t} \right)$ and we rewrite Equation 22 as

$$u_t(\boldsymbol{x}_t | \boldsymbol{x}_T) = \frac{a'_t}{a_t} \boldsymbol{x}_t - \frac{b_t}{2} \lambda'_t \boldsymbol{x}_T \tag{23}$$

By applying the reparameterization from Equation 23 to Equation 1, we establish the noise prediction target:

$$\begin{aligned}
\mathcal{L}_{\text{CFM}} &= \mathbb{E}_{t, \boldsymbol{x}_t \sim p_t(\boldsymbol{x}_t | \boldsymbol{x}_T), \boldsymbol{x}_T \sim p_T(\boldsymbol{x}_T)} \| v_\theta(\boldsymbol{x}_t, t) - u_t(\boldsymbol{x}_t | \boldsymbol{x}_T) \|_2^2 \\
&= \mathbb{E}_{t, \boldsymbol{x}_t \sim p_t(\boldsymbol{x}_t | \boldsymbol{x}_T), \boldsymbol{x}_T \sim p_T(\boldsymbol{x}_T)} \left\| v_\theta(\boldsymbol{x}_t, t) - \frac{a'_t}{a_t} \boldsymbol{x}_t + \frac{b_t}{2} \lambda'_t \boldsymbol{x}_T \right\|_2^2 \\
&= \mathbb{E}_{t, \boldsymbol{x}_t \sim p_t(\boldsymbol{x}_t | \boldsymbol{x}_T), \boldsymbol{x}_T \sim p_T(\boldsymbol{x}_T)} \left( -\frac{b_t}{2} \lambda'_t \right)^2 \| \epsilon_\theta(\boldsymbol{x}_t, t) - \epsilon \|_2^2,
\end{aligned} \tag{24}$$

where $\epsilon_\theta := \frac{2}{\lambda'_t b_t} \left( \frac{a'_t}{a_t} \boldsymbol{x}_t - v_\theta \right)$ and $\epsilon = \boldsymbol{x}_T$. Importantly, incorporating time-varying weights does not affect the objective's optimal solution. This flexibility allows the construction of alternative loss functions that maintain correctness but influence training behavior. For comparative analysis across methodologies (traditional diffusion included), we define the unified objective as Equation 2.

## C. Details of the Primary Derivation

In this section, we present a detailed derivation of our proposed method. Following Diffusion-DPO, we define the reward on the whole chain:

$$r(\boldsymbol{x}_0, \boldsymbol{c}) = \mathbb{E}_{p_\theta(\boldsymbol{x}_{1:T}|\boldsymbol{x}_0, \boldsymbol{c})}[r(\boldsymbol{x}_{0:T}, \boldsymbol{c})]. \tag{25}$$

We begin with the objective function of RLHF:

$$\begin{aligned}
&\max_{p_\theta} \mathbb{E}_{\boldsymbol{x}_0 \sim p_\theta(\boldsymbol{x}_0|\boldsymbol{c})}[r(\boldsymbol{x}_0, \boldsymbol{c})]/\beta - \mathbb{D}_{\mathrm{KL}}[p_\theta(\boldsymbol{x}_0|\boldsymbol{c})||p_{\mathrm{ref}}(\boldsymbol{x}_0|\boldsymbol{c})] \\
&= \min_{p_\theta^{\boldsymbol{c}}} -\mathbb{E}_{\boldsymbol{x}_0 \sim p_\theta^{\boldsymbol{c}}(\boldsymbol{x}_0)}[r(\boldsymbol{x}_0, \boldsymbol{c})]/\beta + \mathbb{D}_{\mathrm{KL}}[p_\theta^{\boldsymbol{c}}(\boldsymbol{x}_0)||p_{\mathrm{ref}}^{\boldsymbol{c}}(\boldsymbol{x}_0)] \\
&\leq \min_{p_\theta^{\boldsymbol{c}}} -\mathbb{E}_{\boldsymbol{x}_0 \sim p_\theta^{\boldsymbol{c}}(\boldsymbol{x}_0)}[r(\boldsymbol{x}_0, \boldsymbol{c})]/\beta + \mathbb{D}_{\mathrm{KL}}[p_\theta^{\boldsymbol{c}}(\boldsymbol{x}_{0:T})||p_{\mathrm{ref}}^{\boldsymbol{c}}(\boldsymbol{x}_{0:T})] \\
&= \min_{p_\theta^{\boldsymbol{c}}} -\mathbb{E}_{p_\theta^{\boldsymbol{c}}(\boldsymbol{x}_{0:T})}[r^{\boldsymbol{c}}(\boldsymbol{x}_{0:T})]/\beta + \mathbb{D}_{\mathrm{KL}}[p_\theta^{\boldsymbol{c}}(\boldsymbol{x}_{0:T})||p_{\mathrm{ref}}^{\boldsymbol{c}}(\boldsymbol{x}_{0:T})] \\
&= \min_{p_\theta^{\boldsymbol{c}}} \mathbb{E}_{p_\theta^{\boldsymbol{c}}(\boldsymbol{x}_{0:T})} \left( \log \frac{p_\theta^{\boldsymbol{c}}(\boldsymbol{x}_{0:T})}{p_{\mathrm{ref}}^{\boldsymbol{c}}(\boldsymbol{x}_{0:T}) \exp(r_t^{\boldsymbol{c}}(\boldsymbol{x}_{0:T})/\beta)/Z(\boldsymbol{c})} - \log Z(\boldsymbol{c}) \right) \\
&= \min_{p_\theta^{\boldsymbol{c}}} \mathbb{D}_{\mathrm{KL}}(p_\theta^{\boldsymbol{c}}(\boldsymbol{x}_{0:T})||p_{\mathrm{ref}}^{\boldsymbol{c}}(\boldsymbol{x}_{0:T}) \exp(r^{\boldsymbol{c}}(\boldsymbol{x}_{0:T})/\beta)/Z(\boldsymbol{c}))
\end{aligned} \tag{26}$$

where $Z(\boldsymbol{c}) = \sum_{\boldsymbol{x}_0} p_{\mathrm{ref}}^{\boldsymbol{c}}(\boldsymbol{x}_{0:T}) \exp(r(\boldsymbol{x}_0, \boldsymbol{c})/\beta)$. The optimization problem defined in Equation 26 has a unique closed-form solution for the conditional distribution:

$$p_{\theta*}^{\boldsymbol{c}}(\boldsymbol{x}_{0:T}) = p_{\mathrm{ref}}^{\boldsymbol{c}}(\boldsymbol{x}_{0:T}) \exp(r^{\boldsymbol{c}}(\boldsymbol{x}_{0:T})/\beta)/Z(\boldsymbol{c}). \tag{27}$$

A direct transformation of Equation (20) yields the solution for the joint reward function:

$$r^{\boldsymbol{c}}(\boldsymbol{x}_{0:T}) = \beta \log \frac{p_{\theta*}^{\boldsymbol{c}}(\boldsymbol{x}_{0:T})}{p_{\mathrm{ref}}^{\boldsymbol{c}}(\boldsymbol{x}_{0:T})} + \beta \log Z(\boldsymbol{c}). \tag{28}$$

Based on Equation 25, we obtain the following expression for the initial reward:

$$r(\boldsymbol{x}_0, \boldsymbol{c}) = \beta \mathbb{E}_{p_\theta^{\boldsymbol{c}}(\boldsymbol{x}_{1:T}|\boldsymbol{x}_0)} \left[ \log \frac{p_{\theta*}^{\boldsymbol{c}}(\boldsymbol{x}_{0:T})}{p_{\mathrm{ref}}^{\boldsymbol{c}}(\boldsymbol{x}_{0:T})} \right] + \beta \log Z(\boldsymbol{c}) \tag{29}$$

Through reward reparameterization and its incorporation into the Bradley-Terry model's maximum likelihood framework, we observe cancellation of the pairwise partition functions. This leads to a tractable maximum likelihood objective for the diffusion model, whose instance-specific formulation of Diffusion-DPO is :

$$\mathcal{L}_{\mathrm{DPO-Diffusion}}(\theta) = -\mathbb{E}_{(\boldsymbol{c}, \boldsymbol{x}_0^w, \boldsymbol{x}_0^l) \sim \mathcal{D}} \log \sigma \left( \beta \mathbb{E}_{\substack{\boldsymbol{x}_{1:T}^w \sim p_\theta^{\boldsymbol{c}}(\boldsymbol{x}_{1:T}^w|\boldsymbol{x}_0^w) \\ \boldsymbol{x}_{1:T}^l \sim p_\theta^{\boldsymbol{c}}(\boldsymbol{x}_{1:T}^l|\boldsymbol{x}_0^l)}} \left[ \log \frac{p_\theta^{\boldsymbol{c}}(\boldsymbol{x}_{0:T}^w)}{p_{\mathrm{ref}}^{\boldsymbol{c}}(\boldsymbol{x}_{0:T}^w)} - \log \frac{p_\theta^{\boldsymbol{c}}(\boldsymbol{x}_{0:T}^l)}{p_{\mathrm{ref}}^{\boldsymbol{c}}(\boldsymbol{x}_{0:T}^l)} \right] \right). \tag{30}$$

According to the conditional probability formula $p_\theta^{\boldsymbol{c}}(\boldsymbol{x}_{1:T}|\boldsymbol{x}_0) = p_\theta^{\boldsymbol{c}}(\boldsymbol{x}_T|\boldsymbol{x}_0)p_\theta^{\boldsymbol{c}}(\boldsymbol{x}_{1:T-1}|\boldsymbol{x}_0, \boldsymbol{x}_T)$, we can derive that:

$$\mathcal{L}(\theta) = -\mathbb{E}_{\mathcal{D}} \log \sigma \left( \beta \mathbb{E}_{\substack{\boldsymbol{x}_T^w \sim p_\theta(\boldsymbol{x}_T^w|\boldsymbol{x}_0^w) \\ \boldsymbol{x}_T^l \sim p_\theta(\boldsymbol{x}_T^l|\boldsymbol{x}_0^l)}} \mathbb{E}_{\substack{\boldsymbol{x}_{1:T-1}^w \sim p_\theta^{\boldsymbol{c}}(\boldsymbol{x}_{1:T-1}^w|\boldsymbol{x}_0^w, \boldsymbol{x}_T^w) \\ \boldsymbol{x}_{1:T-1}^l \sim p_\theta^{\boldsymbol{c}}(\boldsymbol{x}_{1:T-1}^l|\boldsymbol{x}_0^l, \boldsymbol{x}_T^l)}} \left[ \log \frac{p_\theta^{\boldsymbol{c}}(\boldsymbol{x}_{0:T}^w)}{p_{\mathrm{ref}}^{\boldsymbol{c}}(\boldsymbol{x}_{0:T}^w)} - \log \frac{p_\theta^{\boldsymbol{c}}(\boldsymbol{x}_{0:T}^l)}{p_{\mathrm{ref}}^{\boldsymbol{c}}(\boldsymbol{x}_{0:T}^l)} \right] \right). \tag{31}$$

Given $\boldsymbol{x}_T^*$, $p_\theta^{\boldsymbol{c}}(\boldsymbol{x}_{1:T-1}^*|\boldsymbol{x}_0^*, \boldsymbol{x}_T^*)$ becomes tractable if we estimate it using $p_\theta^{\boldsymbol{c}}(\boldsymbol{x}_{1:T-1}^*|\boldsymbol{x}_T^*)$, though this approach is evidently resource-intensive. Leveraging the inherent straightness of rectified flow's sampling trajectories, we can instead estimate

$p_\theta^{\mathbf{c}}(\boldsymbol{x}_{1:T-1}^*|\boldsymbol{x}_0^*, \boldsymbol{x}_T^*)$ using an interpolation-based approximation $q(\boldsymbol{x}_{1:T-1}^*|\boldsymbol{x}_0^*, \boldsymbol{x}_T^*)$.

$$= -\mathbb{E}_{\mathcal{D}} \log \sigma \left( \beta \mathbb{E}_{\substack{\boldsymbol{x}_T^w \sim p_\theta(\boldsymbol{x}_T^w|\boldsymbol{x}_0^w) \\ \boldsymbol{x}_T^l \sim p_\theta(\boldsymbol{x}_T^l|\boldsymbol{x}_0^l)}} \mathbb{E}_{\substack{\boldsymbol{x}_{1:T-1}^w \sim q(\boldsymbol{x}_{1:T-1}^w|\boldsymbol{x}_0^w, \boldsymbol{x}_T^w) \\ \boldsymbol{x}_{1:T-1}^l \sim q(\boldsymbol{x}_{1:T-1}^l|\boldsymbol{x}_0^l, \boldsymbol{x}_T^l)}} \left[ \log \frac{p_\theta^{\mathbf{c}}(\boldsymbol{x}_{0:T}^w)}{p_{\mathrm{ref}}^{\mathbf{c}}(\boldsymbol{x}_{0:T}^w)} - \log \frac{p_\theta^{\mathbf{c}}(\boldsymbol{x}_{0:T}^l)}{p_{\mathrm{ref}}^{\mathbf{c}}(\boldsymbol{x}_{0:T}^l)} \right] \right)$$

$$= -\mathbb{E}_{\mathcal{D}} \log \sigma \left( \beta \mathbb{E}_{\substack{\boldsymbol{x}_T^w \sim p_\theta(\boldsymbol{x}_T^w|\boldsymbol{x}_0^w) \\ \boldsymbol{x}_T^l \sim p_\theta(\boldsymbol{x}_T^l|\boldsymbol{x}_0^l)}} \mathbb{E}_{\substack{\boldsymbol{x}_{1:T-1}^w \sim q(\boldsymbol{x}_{1:T-1}^w|\boldsymbol{x}_0^w, \boldsymbol{x}_T^w) \\ \boldsymbol{x}_{1:T-1}^l \sim q(\boldsymbol{x}_{1:T-1}^l|\boldsymbol{x}_0^l, \boldsymbol{x}_T^l)}} \left[ \sum_{t=1}^{T} \log \frac{p_\theta^{\mathbf{c}}(\boldsymbol{x}_{t-1}^w|\boldsymbol{x}_t^w)}{p_{\mathrm{ref}}^{\mathbf{c}}(\boldsymbol{x}_{t-1}^w|\boldsymbol{x}_t^w)} - \log \frac{p_\theta^{\mathbf{c}}(\boldsymbol{x}_{t-1}^l|\boldsymbol{x}_t^l)}{p_{\mathrm{ref}}^{\mathbf{c}}(\boldsymbol{x}_{t-1}^l|\boldsymbol{x}_t^l)} \right] \right)$$

$$= -\mathbb{E}_{\mathcal{D}} \log \sigma \left( \beta \mathbb{E}_{\substack{\boldsymbol{x}_T^w \sim p_\theta(\boldsymbol{x}_T^w|\boldsymbol{x}_0^w) \\ \boldsymbol{x}_T^l \sim p_\theta(\boldsymbol{x}_T^l|\boldsymbol{x}_0^l)}} \mathbb{E}_{\substack{\boldsymbol{x}_{1:T-1}^w \sim q(\boldsymbol{x}_{1:T-1}^w|\boldsymbol{x}_0^w, \boldsymbol{x}_T^w) \\ \boldsymbol{x}_{1:T-1}^l \sim q(\boldsymbol{x}_{1:T-1}^l|\boldsymbol{x}_0^l, \boldsymbol{x}_T^l)}} T\mathbb{E}_t \left[ \log \frac{p_\theta^{\mathbf{c}}(\boldsymbol{x}_{t-1}^w|\boldsymbol{x}_t^w)}{p_{\mathrm{ref}}^{\mathbf{c}}(\boldsymbol{x}_{t-1}^w|\boldsymbol{x}_t^w)} - \log \frac{p_\theta^{\mathbf{c}}(\boldsymbol{x}_{t-1}^l|\boldsymbol{x}_t^l)}{p_{\mathrm{ref}}^{\mathbf{c}}(\boldsymbol{x}_{t-1}^l|\boldsymbol{x}_t^l)} \right] \right)$$

$$= -\mathbb{E}_{\mathcal{D}} \log \sigma \left( \beta T\mathbb{E}_t \mathbb{E}_{\substack{\boldsymbol{x}_T^w \sim p_\theta(\boldsymbol{x}_T^w|\boldsymbol{x}_0^w) \\ \boldsymbol{x}_T^l \sim p_\theta(\boldsymbol{x}_T^l|\boldsymbol{x}_0^l)}} \mathbb{E}_{\substack{\boldsymbol{x}_{t-1,t}^w \sim q(\boldsymbol{x}_{t-1,t}^w|\boldsymbol{x}_0^w, \boldsymbol{x}_T^w) \\ \boldsymbol{x}_{t-1,t}^l \sim q(\boldsymbol{x}_{t-1,t}^l|\boldsymbol{x}_0^l, \boldsymbol{x}_T^l)}} \left[ \log \frac{p_\theta^{\mathbf{c}}(\boldsymbol{x}_{t-1}^w|\boldsymbol{x}_t^w)}{p_{\mathrm{ref}}^{\mathbf{c}}(\boldsymbol{x}_{t-1}^w|\boldsymbol{x}_t^w)} - \log \frac{p_\theta^{\mathbf{c}}(\boldsymbol{x}_{t-1}^l|\boldsymbol{x}_t^l)}{p_{\mathrm{ref}}^{\mathbf{c}}(\boldsymbol{x}_{t-1}^l|\boldsymbol{x}_t^l)} \right] \right) \quad (32)$$

$$= -\mathbb{E}_{\mathcal{D}} \log \sigma \left( \beta T\mathbb{E}_t \mathbb{E}_{\boldsymbol{x}_T^w \sim p_\theta(\boldsymbol{x}_T^w|\boldsymbol{x}_0^w), \boldsymbol{x}_T^l \sim p_\theta(\boldsymbol{x}_T^l|\boldsymbol{x}_0^l)} \mathbb{E}_{\boldsymbol{x}_t^w \sim q(\boldsymbol{x}_t^w|\boldsymbol{x}_0^w, \boldsymbol{x}_T^w), \boldsymbol{x}_t^l \sim q(\boldsymbol{x}_t^l|\boldsymbol{x}_0^l, \boldsymbol{x}_T^l)} \right.$$

$$\left. \mathbb{E}_{\boldsymbol{x}_t^w \sim q(\boldsymbol{x}_{t-1}^w|\boldsymbol{x}_0^w, \boldsymbol{x}_t^w, \boldsymbol{x}_T^w), \boldsymbol{x}_{t-1}^l \sim q(\boldsymbol{x}_{t-1}^l|\boldsymbol{x}_0^l, \boldsymbol{x}_t^l, \boldsymbol{x}_T^l)} \left[ \log \frac{p_\theta^{\mathbf{c}}(\boldsymbol{x}_{t-1}^w|\boldsymbol{x}_t^w)}{p_{\mathrm{ref}}^{\mathbf{c}}(\boldsymbol{x}_{t-1}^w|\boldsymbol{x}_t^w)} - \log \frac{p_\theta^{\mathbf{c}}(\boldsymbol{x}_{t-1}^l|\boldsymbol{x}_t^l)}{p_{\mathrm{ref}}^{\mathbf{c}}(\boldsymbol{x}_{t-1}^l|\boldsymbol{x}_t^l)} \right] \right)$$

$$= -\mathbb{E}_{\mathcal{D}} \log \sigma \left( \beta T\mathbb{E}_t \mathbb{E}_{\boldsymbol{x}_T^w \sim p_\theta(\boldsymbol{x}_T^w|\boldsymbol{x}_0^w), \boldsymbol{x}_T^l \sim p_\theta(\boldsymbol{x}_T^l|\boldsymbol{x}_0^l)} \mathbb{E}_{\boldsymbol{x}_t^w \sim q(\boldsymbol{x}_t^w|\boldsymbol{x}_0^w, \boldsymbol{x}_T^w), \boldsymbol{x}_t^l \sim q(\boldsymbol{x}_t^l|\boldsymbol{x}_0^l, \boldsymbol{x}_T^l)} \right.$$

$$\left. \mathbb{E}_{\boldsymbol{x}_t^w \sim q(\boldsymbol{x}_{t-1}^w|\boldsymbol{x}_0^w, \boldsymbol{x}_t^w), \boldsymbol{x}_{t-1}^l \sim q(\boldsymbol{x}_{t-1}^l|\boldsymbol{x}_0^l, \boldsymbol{x}_t^l)} \left[ \log \frac{p_\theta^{\mathbf{c}}(\boldsymbol{x}_{t-1}^w|\boldsymbol{x}_t^w)}{p_{\mathrm{ref}}^{\mathbf{c}}(\boldsymbol{x}_{t-1}^w|\boldsymbol{x}_t^w)} - \log \frac{p_\theta^{\mathbf{c}}(\boldsymbol{x}_{t-1}^l|\boldsymbol{x}_t^l)}{p_{\mathrm{ref}}^{\mathbf{c}}(\boldsymbol{x}_{t-1}^l|\boldsymbol{x}_t^l)} \right] \right).$$

According to Jensen's inequality, we have:

$$\mathcal{L}(\theta) \leq - \mathbb{E}_{\mathcal{D},t} \mathbb{E}_{\boldsymbol{x}_T^w \sim p_\theta(\boldsymbol{x}_T^w|\boldsymbol{x}_0^w), \boldsymbol{x}_T^l \sim p_\theta(\boldsymbol{x}_T^l|\boldsymbol{x}_0^l)} \mathbb{E}_{\boldsymbol{x}_t^w \sim q(\boldsymbol{x}_t^w|\boldsymbol{x}_0^w, \boldsymbol{x}_T^w), \boldsymbol{x}_t^l \sim q(\boldsymbol{x}_t^l|\boldsymbol{x}_0^l, \boldsymbol{x}_T^l)} \log \sigma \Bigg($$

$$\beta T \mathbb{E}_{\boldsymbol{x}_t^w \sim q(\boldsymbol{x}_{t-1}^w|\boldsymbol{x}_0^w, \boldsymbol{x}_t^w), \boldsymbol{x}_{t-1}^l \sim q(\boldsymbol{x}_{t-1}^l|\boldsymbol{x}_0^l, \boldsymbol{x}_t^l)} \left[ \log \frac{p_\theta^{\mathbf{c}}(\boldsymbol{x}_{t-1}^w|\boldsymbol{x}_t^w)}{p_{\mathrm{ref}}^{\mathbf{c}}(\boldsymbol{x}_{t-1}^w|\boldsymbol{x}_t^w)} - \log \frac{p_\theta^{\mathbf{c}}(\boldsymbol{x}_{t-1}^l|\boldsymbol{x}_t^l)}{p_{\mathrm{ref}}^{\mathbf{c}}(\boldsymbol{x}_{t-1}^l|\boldsymbol{x}_t^l)} \right] \Bigg)$$

$$= - \mathbb{E}_{\mathcal{D},t} \mathbb{E}_{\boldsymbol{x}_T^w \sim p_\theta(\boldsymbol{x}_T^w|\boldsymbol{x}_0^w), \boldsymbol{x}_T^l \sim p_\theta(\boldsymbol{x}_T^l|\boldsymbol{x}_0^l)} \mathbb{E}_{\boldsymbol{x}_t^w \sim q(\boldsymbol{x}_t^w|\boldsymbol{x}_0^w, \boldsymbol{x}_T^w), \boldsymbol{x}_t^l \sim q(\boldsymbol{x}_t^l|\boldsymbol{x}_0^l, \boldsymbol{x}_T^l)} \log \sigma \Bigg( \quad (33)$$

$$- \beta T \Bigg[ \mathbb{D}_{\mathrm{KL}}(q(\boldsymbol{x}_{t-1}^w|\boldsymbol{x}_{0,t}^w)||p_\theta^{\mathbf{c}}(\boldsymbol{x}_{t-1}^w|\boldsymbol{x}_t^w)) - \mathbb{D}_{\mathrm{KL}}(q(\boldsymbol{x}_{t-1}^w|\boldsymbol{x}_{0,t}^w)||p_{\mathrm{ref}}^{\mathbf{c}}(\boldsymbol{x}_{t-1}^w|\boldsymbol{x}_t^w))$$

$$- \mathbb{D}_{\mathrm{KL}}(q(\boldsymbol{x}_{t-1}^l|\boldsymbol{x}_{0,t}^l)||p_\theta^{\mathbf{c}}(\boldsymbol{x}_{t-1}^l|\boldsymbol{x}_t^l)) - \mathbb{D}_{\mathrm{KL}}(q(\boldsymbol{x}_{t-1}^l|\boldsymbol{x}_{0,t}^l)||p_{\mathrm{ref}}^{\mathbf{c}}(\boldsymbol{x}_{t-1}^l|\boldsymbol{x}_t^l)) \Bigg] \Bigg)$$

Through parameterization of the Rectified Flow reverse process, the aforementioned loss simplifies to:

$$\mathcal{L}_{\mathrm{PNAPO}}(\theta) = -\mathbb{E}_{(\boldsymbol{c}, \boldsymbol{x}_0^w, \boldsymbol{x}_0^l, \boldsymbol{x}_T^w, \boldsymbol{x}_T^l) \sim \mathcal{D}_{\mathrm{PNAPO}}, t} \log \sigma(-\beta(\boldsymbol{s}_\theta^t(\boldsymbol{x}_0^w, \boldsymbol{x}_T^w, \boldsymbol{c}) - \boldsymbol{s}_\theta^t(\boldsymbol{x}_0^l, \boldsymbol{x}_T^l, \boldsymbol{c}))) \quad (34)$$

where $t \sim \mathcal{U}(0, T)$ and we define the interpolation-based score function $\boldsymbol{s}_\theta^t$ as:

$$\boldsymbol{s}_\theta^t(\boldsymbol{x}_0^*, \boldsymbol{x}_T^*, \boldsymbol{c}) = \|(\boldsymbol{x}_T^* - \boldsymbol{x}_0^*) - v_\theta(\boldsymbol{x}_t^*, t, \boldsymbol{c})\|_2^2 - \|(\boldsymbol{x}_T^* - \boldsymbol{x}_0^*) - v_{\mathrm{ref}}(\boldsymbol{x}_t^*, t, \boldsymbol{c})\|_2^2, \quad (35)$$

where $\boldsymbol{x}_t^* = (1 - t)\boldsymbol{x}_0^* + t\boldsymbol{x}_T^*$.

**Why PNAPO is better than Diffusion-DPO?** Notably, while Diffusion-DPO employs the forward process $q(\boldsymbol{x}_{1:T}|\boldsymbol{x}_0)$ to estimate the reverse process $p_\theta(\boldsymbol{x}_{1:T}|\boldsymbol{x}_0)$, our method utilizes $p_\theta(\boldsymbol{x}_T|\boldsymbol{x}_0)q(\boldsymbol{x}_{1:T-1}|\boldsymbol{x}_0, \boldsymbol{x}_T)$ for estimation. This approximation yields lower error since

$$\mathbb{D}_{\mathrm{KL}}(p_\theta(\boldsymbol{x}_T|\boldsymbol{x}_0)q(\boldsymbol{x}_{1:T-1}|\boldsymbol{x}_0, \boldsymbol{x}_T)||p_\theta(\boldsymbol{x}_{1:T}|\boldsymbol{x}_0)) = \mathbb{D}_{\mathrm{KL}}(q(\boldsymbol{x}_{1:T-1}|\boldsymbol{x}_0, \boldsymbol{x}_T)||p_\theta(\boldsymbol{x}_{1:T-1}|\boldsymbol{x}_0, \boldsymbol{x}_T))$$
$$\leq \mathbb{D}_{\mathrm{KL}}(q(\boldsymbol{x}_{1:T}|\boldsymbol{x}_0)||p_\theta(\boldsymbol{x}_{1:T}|\boldsymbol{x}_0)). \quad (36)$$

# D. Further Discussion

### D.1. Disscussion of Dataset

In T2I diffusion models, human preference feedback is influenced by multifaceted factors such as image quality, photorealism, artistic style, and cultural context. The inherently subjective nature of these factors, coupled with prevalent noise in datasets, presents challenges for AI systems to learn effectively, thereby underscoring the critical importance of robust preference learning. Furthermore, the inherent diversity and uncertainty of human preferences during T2I diffusion introduce substantial modeling complexity and may lead to potential distributional shifts. Although Diffusion-DB prompt dataset has amassed an extensive collection of text prompts, it exhibits notable sampling bias with substantial text repetition. To mitigate this bias, we employed rigorous data cleansing procedures combined with KNN-based diversity sampling to construct a more balanced textual dataset.

### D.2. Limitations and Future Work

Our current approach is constrained to enhancing model performance exclusively through noise-image pairs generated by the model itself. Specifically, we cannot fine-tune SD3-M using dataset generated by FLUX due to inherent noise distribution discrepancies. Future research directions will focus on: extending our method to online learning paradigms, and developing adaptive parameter optimization strategies. Currently, the text prompts in the Diffusion-DB dataset lack coherence, which may limit their effectiveness in guiding high-quality image generation. To address this, we propose leveraging multimodal large language models (MLLMs) for prompt alignment and refinement. This approach could enhance overall image quality—either by improving the entire dataset or selectively optimizing high-quality samples.

# E. Experiment Details

**HPDv2**    HPDv2 collects human preference data through the "Dreambot" channel on Stable Foundation's Discord server. The dataset comprises 25,205 distinct text prompts used to generate 98,807 images in total. Each text prompt is associated with: Paired image labels indicating users' relative preferences between image pairs. The number of generated images per prompt varies across the dataset. For our experiments, we utilize the test set containing 3,200 text prompts from this collection.

**OPDv1**    The Open Preference Dataset represents a collaborative effort among Hugging Face, Argilla, and the open-source machine learning community. This initiative is strategically designed to empower the open-source ecosystem through the co-creation of impactful datasets for generative AI research. The current release comprises 7,459 meticulously curated text-to-image preference pairs, serving as a benchmark resource for: Comprehensive evaluation of image generation models across diverse semantic categories; Systematic performance assessment through stratified prompts of varying complexity levels; Comparative analysis of model outputs against human perceptual preferences.

### E.1. Additional Implementation details

We configured our pipeline with Detoxify's NSFW score threshold at 0.1 for content filtering, Jaccard similarity threshold at 0.8 for textual redundancy removal, and ViT-H/CLIP embedding cosine similarity threshold at 0.8 for semantic deduplication. For balanced prompt sampling, we performed KNN clustering (K=100) with 200 prompts per cluster. All models were trained on 8 NVIDIA H800 GPUs, with SD3-M using gradient accumulation over 8 steps (batch size=1/GPU) and FLUX using single-step accumulation. To ensure fair evaluation, we maintained consistent sampling parameters across experiments: CFG scale=1, 50 sampling steps, and fixed random seeds. The optimization used a learning rate of $1e^{-6}$ with 500-step linear warmup. For FLUX, lora rank is set to 32.

### E.2. Off-Policy Data Construction

We present a subset of samples generated by FLUX as Figure 6 and Figure 7, with comparative assessments conducted using HPSv2.1 (Human Preference Score v2.1) as the evaluation metric. These visualizations demonstrate both the quality improvements achieved through rectification and the effectiveness of HPSv2.1 in discriminating between sample variations.

# F. Additional Quantitative Results

We additionally sample 3,000 prompts from Diffusion-DB as a test set and present further quantitative results. Experiments demonstrate that our method generates images with superior aesthetics and text alignment compared to various baselines.

*Table 8.* **Additional quantitative comparison with FLUX baselines.** We apply the prompts from the Diffusion-DB test set to compare our model with the existing alignment baseline on the FLUX model. We report the median and mean values of five reward evaluators on the Diffusion-DB test set, retaining five significant figures. In the table, the highest value in each column is highlighted in **bold**. As shown, our model achieves the best results in nearly all reward evaluations.

| Baselines | PickScore | | HPSv2.1 | | ImageReward | | Aesthetic | | CLIP | |
|---|---|---|---|---|---|---|---|---|---|---|
| | Median | Mean | Median | Mean | Median | Mean | Median | Mean | Median | Mean |
| FLUX | 24.701 | 24.707 | 33.716 | 33.429 | 1.787 | 1.602 | 6.286 | 6.274 | 39.067 | 39.243 |
| SFT | 24.723 | 24.727 | 33.929 | 33.658 | 1.80 | 1.650 | 6.339 | 6.340 | 39.316 | 39.369 |
| DPO | 24.751 | 24.746 | 33.901 | 33.754 | 1.805 | 1.677 | 6.311 | 6.315 | 39.405 | 39.374 |
| IPO | 24.739 | 24.728 | 33.953 | 33.703 | 1.803 | 1.663 | 6.348 | 6.362 | 39.351 | 39.366 |
| PNAPO | **24.883** | **24.899** | **34.327** | **34.340** | **1.812** | **1.699** | **6.473** | **6.482** | **39.686** | **39.430** |

*Table 9.* **Additional quantitative comparison with SD3-M baselines.** We apply the prompts from the Diffusion-DB test set to compare our model with the existing alignment baseline on the SD3-M model. We report the median and mean values of five reward evaluators on the Diffusion-DB test set, retaining five significant figures. In the table, the highest value in each column is highlighted in **bold**. As shown, our model achieves the best results in nearly all reward evaluations.

| Baselines | PickScore | | HPSv2.1 | | ImageReward | | Aesthetic | | CLIP | |
|---|---|---|---|---|---|---|---|---|---|---|
| | Median | Mean | Median | Mean | Median | Mean | Median | Mean | Median | Mean |
| SD3-M | 24.076 | 24.079 | 33.850 | 33.485 | 1.784 | 1.581 | 6.145 | 6.132 | 39.737 | 39.714 |
| SFT | 24.141 | 24.164 | 33.686 | 33.307 | 1.824 | 1.640 | 6.014 | 5.998 | 40.100 | 40.104 |
| DPO | 24.193 | 24.207 | 34.126 | 33.892 | 1.831 | 1.683 | 6.087 | 6.072 | 40.312 | 40.288 |
| IPO | 24.172 | 24.188 | 34.058 | 33.847 | 1.829 | 1.672 | 6.071 | 6.063 | 40.251 | 40.229 |
| PNAPO | **24.246** | **24.251** | **34.748** | **34.285** | **1.838** | **1.714** | **6.212** | **6.164** | **40.489** | **40.481** |

# G. Additional Qualitative Results

To offer more comprehensive insights, we present extended qualitative comparisons as Figure 8, highlighting the advantages of our approach.

| Loser | Winner | Loser | Winner |
| --- | --- | --- | --- |

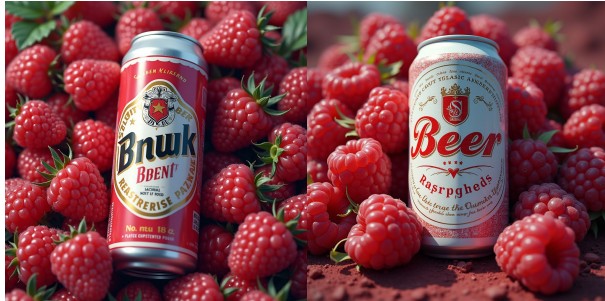 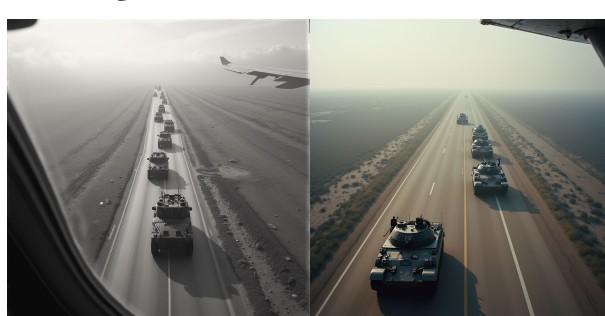

A beautiful beer can surrounded by big oversized overgrown raspberries painted by tom bagshaw, ⋯.

A photo of world war second, troops and armoured vehicles moving, view from plane.

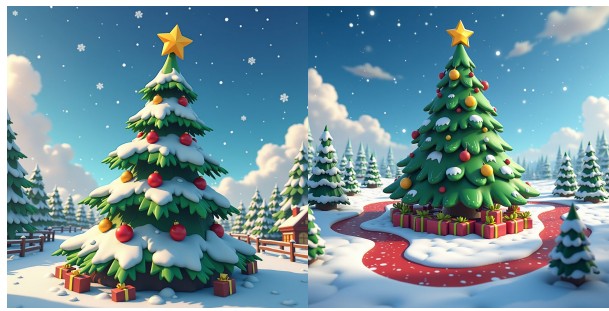 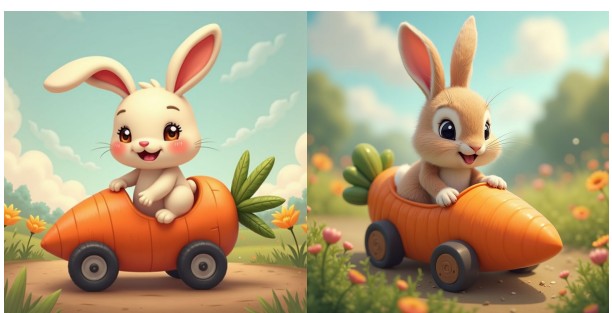

A mario kart track themed around a christmas tree, mario kart 8.

Bunny riding in a carrot car.

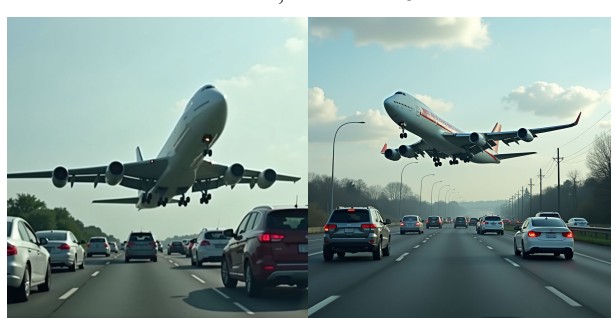 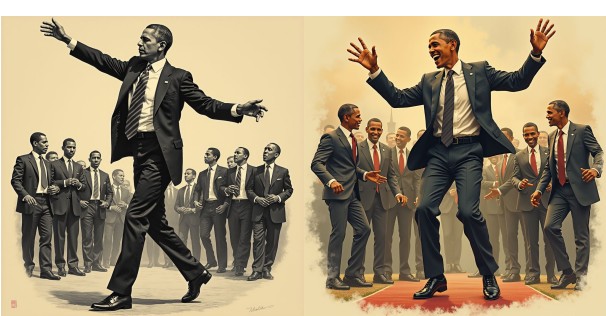

Mobile photo of a 747 plane crashing through traffic on the highway.

Vintage lithograph of barack obama doing a fortnite dance.

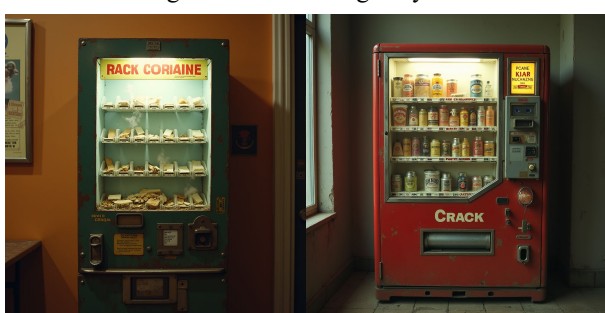 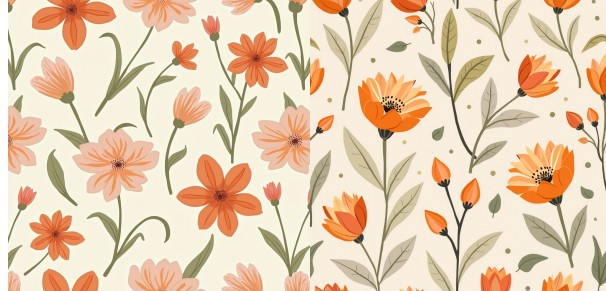

Vending machine for crack cocaine cigarette, soviet propaganda style.

Floral wallpaper with orange pastel colors.

*Figure 6.* Preference Dataset samples generated by FLUX. Both the quality improvements achieved through rectification and the effectiveness of HPSv2.1 in discriminating between sample variations.

| Loser | Winner | Loser | Winner |
|---|---|---|---|

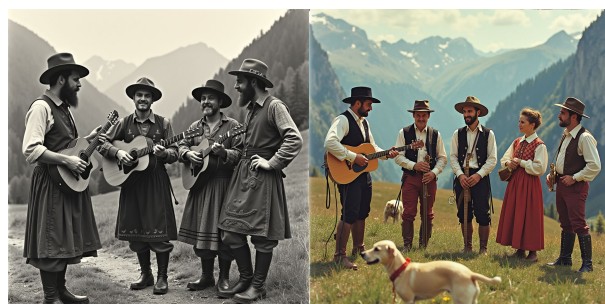 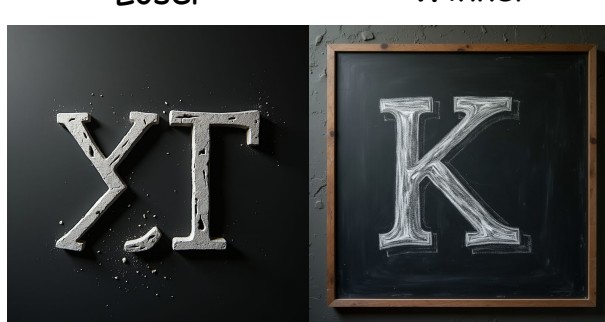

German folklore musicians in traditional costumes in the alps, punk fanzine from 1980, xerox copy machine

Greek alphabet in a blackboard.

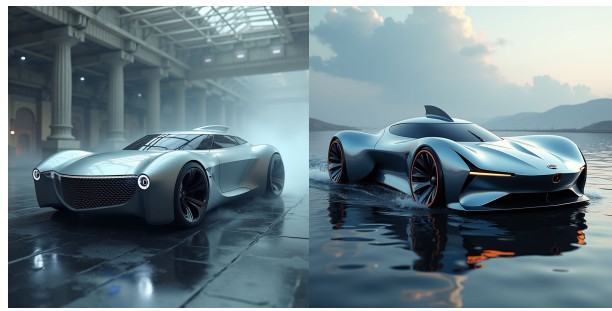 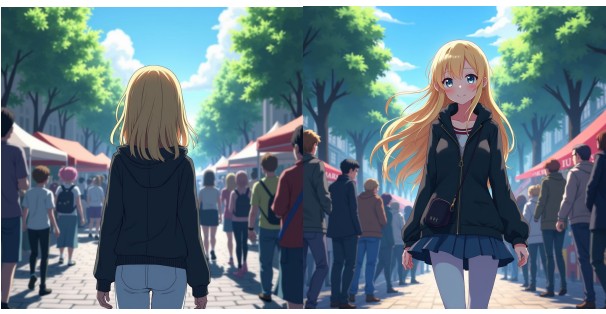

A beautiful concept design of a car that looks like a shark. ⋯

Blonde - haired princess, anime princess, wearing black jacket and white leggings, looking through crowd, ⋯

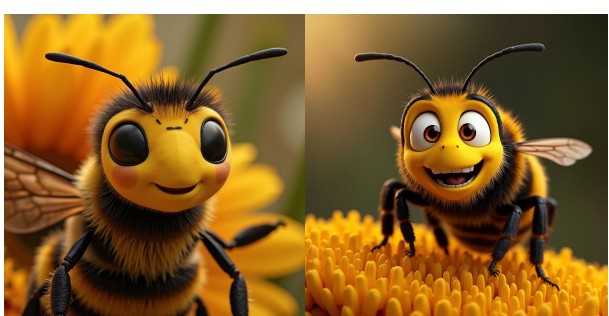 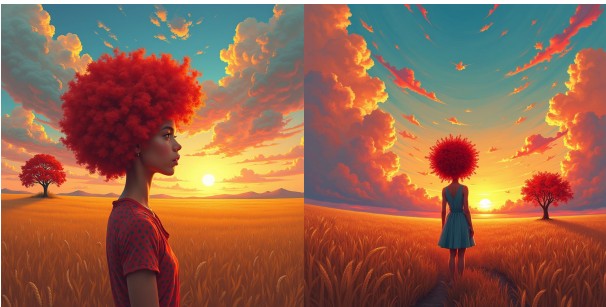

The bee movie with realistic jerry seinfeld head on the bee, in the hive.

Red carnation afro head, full body, girl watching sunset, empty wheat field, surreal photography, colorful clouds, ⋯

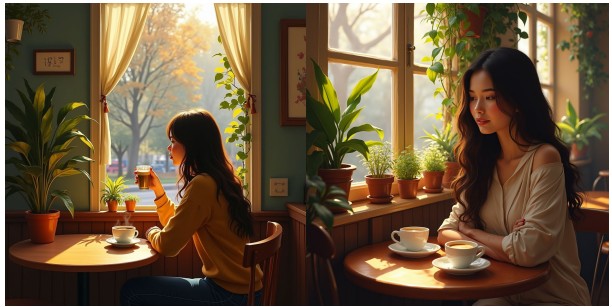 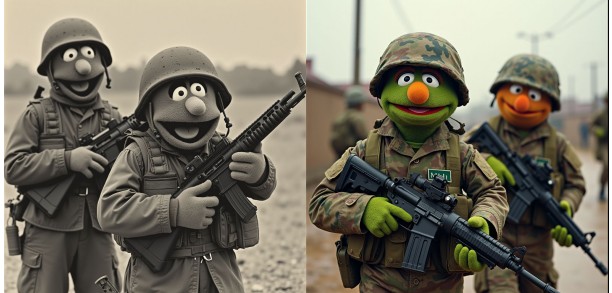

A cozy cute cafe with tables, a big window and plants, a young pretty thin filipino woman with long hair sits ⋯

Muppets special forces unit, associated press photo, war journalism, 150 dpi scan of newsprint

*Figure 7.* Preference Dataset samples generated by FLUX. Both the quality improvements achieved through rectification and the effectiveness of HPSv2.1 in discriminating between sample variations.

FLUX  PNAPO-FLUX  FLUX  PNAPO-FLUX

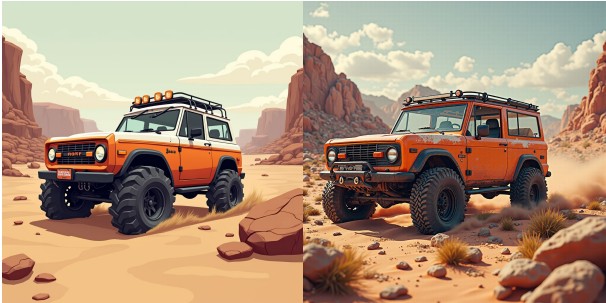 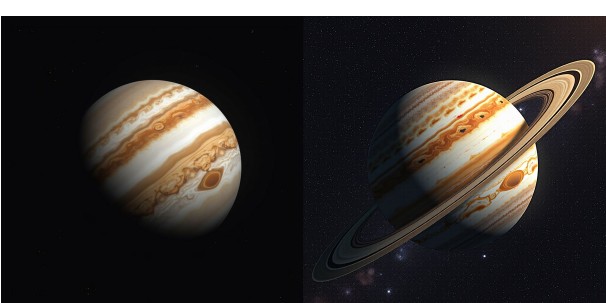

An orange and white off-road vehicle with large tires, driving through a desert landscape with rocks and sand.

The planet Jupiter.

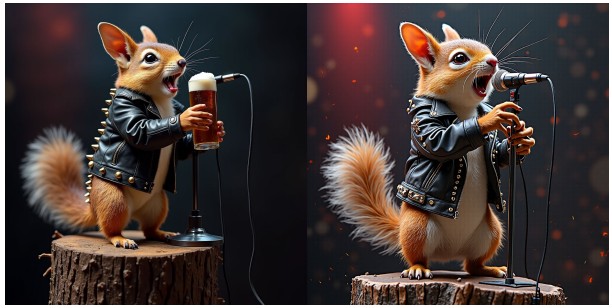 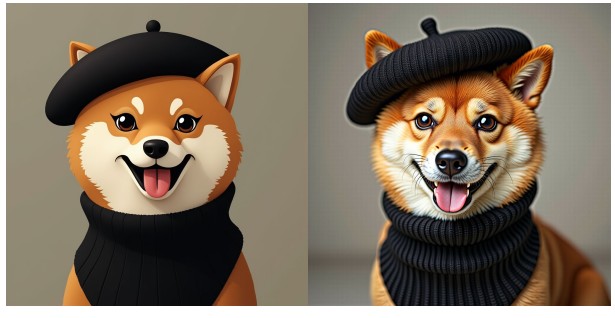

A punk rock squirrel in a studded leather jacket shouting into a microphone ⋯

A shiba inu wearing a beret and black turtleneck

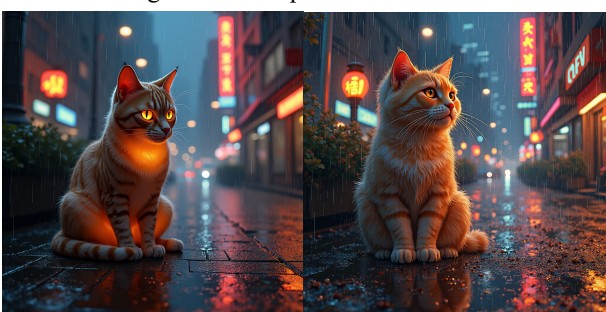 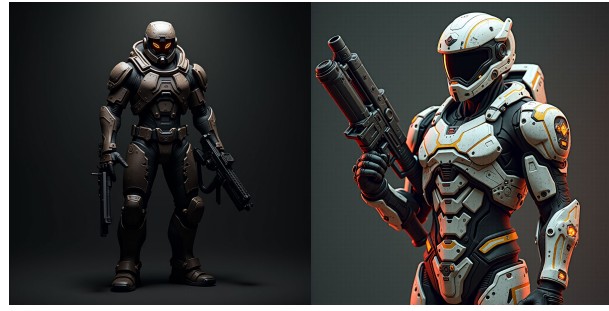

A highly detailed digital rendering of a warm, glowing cat with intricate fur details, sitting alone on a rain ⋯

A futuristic soldier wearing advanced armor and holding a high-tech weapon, standing against a dark background.

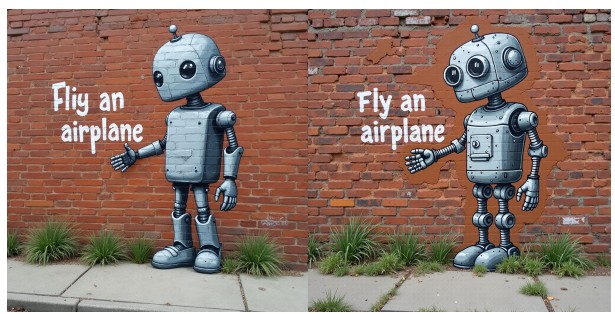 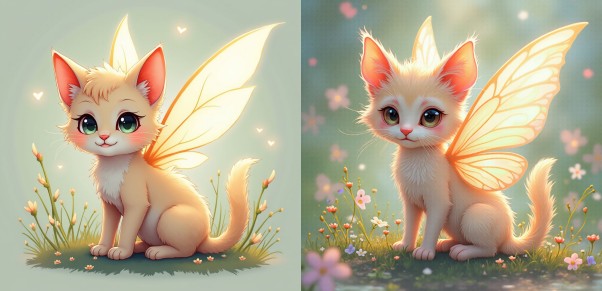

A robot painted ⋯ The words \"Fly an airplane\" are written on the wall.

A petite Druid fairy with cat-like whiskers and ethereal, translucent wings, rendered in vibrant anime style ⋯

*Figure 8.* **Additional Qualitative results.** Compared to FLUX base model, images aligned with our `PNAPO` demonstrate significant improvements in both text-image alignment and aesthetic quality, effectively validating the superiority of our approach.

