# OpenReview forum: "Offline Preference Optimization for Rectified Flow with Noise-Tracked Pairs"
_ICML.cc/2026/Conference — ICML 2026 regular_

### Official Review · Reviewer_j33J · 2026-02-15

**Soundness:** 3
**Presentation:** 2
**Significance:** 3
**Originality:** 3
**Overall Recommendation:** 4
**Confidence:** 4

**Summary:**

This paper is motivated by the insight that diffusion-DPO, when the initial noise is omitted, can suffer from degraded performance. To address this issue, the authors propose the PNAPO method designed for rectified flow. They also introduce a dynamic regularization strategy that adjusts loss weights according to feedback from a reward model. Experimental results, along with ablation studies, demonstrate the effectiveness of the proposed approach.

**Compliance With Llm Reviewing Policy:**

Affirmed.

**Final Justification:**

I think the paper’s motivation is highly insightful. The authors also responded constructively to the concerns I raised, such as reward hacking. It would be even better if the writing were clearer. Overall, considering both the paper’s strengths and the authors’ rebuttal, I would give this paper a weak accept.

**Key Questions For Authors:**

See weakness.

**Limitations:**

See weakness.

**Strengths And Weaknesses:**

**Strengths**
- The motivation is insightful and well worth exploring.
- This paper presents some analysis of the proposed method.
- The experimental results demonstrate the effectiveness of the proposed method.

**Weaknesses**
- First, the presentation of the paper needs significant improvement. The first figure is difficult to understand when initially reading the paper; for example, the meaning of “fixed distance” is unclear at first glance. In addition, some equations (e.g., Eqs. 4 and 8) extend beyond the page margins.
- Although the paper presents a seemingly well-defined motivation, several arguments are not entirely convincing. In particular, the proposed dynamic regularization appears to have limited impact, as suggested by the results in Table 5. Moreover, the visualization in Figure 4 gives the impression that training may suffer from **reward hacking**, resulting in degraded visual quality (for example, the generated cats). One possible explanation is that the dynamic regularization encourages the model to fit certain unusual patterns with disproportionately large weights. In addition, the rationale behind the design of the sample controller, specifically the increasing schedule of f and the decay of \delta, is not sufficiently explained.
- It would be beneficial to include additional visual quality metrics (e.g., VisualQuality-R1 or others in low-level vision tasks) for a more thorough evaluation of the model, since the existing aesthetic metrics are not sufficiently accurate and may fail to reflect overall visual quality.

Overall, I believe this paper could be further improved in terms of writing clarity, evaluation, and visualization. I therefore recommend a weak accept.

---

> ### Author Rebuttal · Authors · 2026-03-30
>
> We sincerely thank you for your positive assessment and helpful suggestions!
>
> ---
>
> **W1:** *Presentation issues: unclear "fixed distance" in Figure 1; equations exceeding margins.*
>
> We commit to addressing all presentation issues in the revision. Specifically:
>
> + **Figure 1**: "Fixed Distance" refers to Diffusion-DPO's use of a fixed $\beta$ applied uniformly to all sample pairs, imposing an identical KL constraint regardless of the quality gap. We will revise the annotation to "Fixed $\beta$ (Uniform KL)" and add an explanatory sentence in the caption contrasting it with PNAPO's dynamic $\beta(\delta_r, n)$.
> + **Equations**: We have identified the overflowing equations (Eq. 4, 8) and will reformat them using multi-line `aligned` environments. We will conduct a full pass to ensure all equations respect column margins.
>
> ---
>
> **W2:** *Dynamic regularization appears to have limited impact (Table 5); Figure 4 suggests reward hacking; design rationale for f and g is insufficiently explained.*
>
> This concern raises three sub-issues, which we address in turn.
>
> **(a) Interpreting Table 5**
>
> We appreciate the reviewer's scrutiny, but we believe the numbers in Table 5 should be interpreted in light of each component's distinct functional role rather than absolute score improvement alone. $f(\delta_r)$ and $g(n)$ operate along orthogonal axes: $f(\delta_r)$ improves **learning efficiency** by upweighting sample pairs with high-confidence preference signals, prioritizing informative training examples; $g(n)$ ensures **training stability** by reducing KL penalty strength in later training stages, preventing the model from being excessively pulled back toward the reference model. Their effects are complementary: $f(\delta_r)$ alone improves HPSv2.1 by +0.58 and Aesthetic by +0.067; $g(n)$ alone improves HPSv2.1 by +0.05 and Aesthetic by +0.035. The combined $\beta(\delta_r, n)$ improves HPSv2.1 by +0.63 and Aesthetic by +0.081 and achieves the best results across all metrics, confirming the two components address orthogonal aspects of training dynamics.
>
> **(b) Abnormal cat features in Figure 4**
>
> We respectfully argue this is not reward hacking but a reflection of **reward model imperfection** (reward model bias issue). HPS v2.1 primarily captures holistic aesthetics, and the cat features in Figure 4 simply reflect the model being guided toward this reward signal. This differs from reward hacking: the model genuinely improves on dimensions HPS v2.1 measures well (confirmed by gains across multiple independent metrics in Table 2), and since $\delta_r \geq 0$ by definition, the range of $f(\delta_r)$ is strictly:
>
> $f(\delta_r) = 2\sigma(\delta_r) - 1 \in [0,\ 1)$, The upper bound is strictly less than 1. The issue is reward-model-specific, not method-specific, motivating future work on multi-dimensional reward signals.
>
> **(c) Insufficient justification for $f$ and $g$**
>
> We acknowledge that the motivation for these design choices was presented too briefly in the original text. We provide a more complete justification here.
>
> For $f(\delta_r) = 2\sigma(\delta_r) - 1$: a larger quality gap $\delta_r$ between winner and loser indicates a more reliable preference annotation and thus a higher-confidence learning signal. When $\delta_r \to 0$, $f \to 0$, and the model effectively ignores near-indistinguishable noisy pairs; when $\delta_r \to \infty$, $f \to 1$, and the model applies full optimization force to pairs with unambiguous preference. The sigmoid form ensures a smooth transition throughout, avoiding gradient discontinuities that would arise from hard thresholds. This design is philosophically aligned with curriculum learning, where sample weights are dynamically adjusted according to signal reliability.
>
> For the cosine decay of $g(n)$: early in training, the model deviates little from the reference model, and a larger $\beta$ is beneficial for driving meaningful parameter updates. In later stages, the model has already diverged substantially; maintaining a high $\beta$ would cause the KL penalty to persistently pull the model back toward the reference, inducing an "alignment-forgetting" oscillation. The decay of $g(n)$ from 1 to 0.5 represents a principled trade-off between continued learning and preventing excessive divergence. We will incorporate these explanations into the main text.
>
> ---
>
> **W3:** *Additional visual quality metrics such as VisualQuality-R1*
>
> Valuable suggestion! We evaluated the models on the HPDv2 test set using VisualQuality-R1 (on a 1-5 scale). Ties (identical scores) were excluded from the win rate calculation. The results are as follows:
>
> Supp-Table 1: VisualQuality-R1 win-rate comparison on SD3-M:
>
> |SD3-M|VisualQuality-R1|
> |-|-|
> |PNAPO vs Pretrain|60.1%|
> |PNAPO vs DPO| 57.6%|
> |PNAPO vs IPO|57.0%|
>
> Supp-Table 2: VisualQuality-R1 win-rate comparison on FLUX:
>
> |FLUX|VisualQuality-R1|
> |-|-|
> |PNAPO vs Pretrain|59.4%|
> |PNAPO vs DPO|56.8%|
> |PNAPO vs IPO|57.2%|

---

> > ### Author Rebuttal · Reviewer_j33J · 2026-04-01
> >
> > Thanks to the author for the detailed rebuttal. My concerns have been satisfactorily addressed, and I will maintain my rating.

---

> > > ### Author Response · Authors · 2026-04-01
> > >
> > > Dear Reviewer j33J,
> > >
> > > Thank you for your time and for reviewing our rebuttal. We are very glad to hear that our response has satisfactorily addressed your concerns. We deeply appreciate your constructive feedback and your support for our work!
> > >
> > > Best regards,
> > >
> > > Authors

---

### Official Review · Reviewer_qGiQ · 2026-03-06

**Soundness:** 3
**Presentation:** 3
**Significance:** 3
**Originality:** 3
**Overall Recommendation:** 5
**Confidence:** 4

**Summary:**

This paper studies offline preference optimization for rectified flow (RF) text-to-image models. The main observation is that standard preference datasets store only the final winner/loser images, which may be insufficient for RF models because generation is indexed by a specific prior noise and follows a nearly straight trajectory. Based on this, the paper proposes Prior Noise-Aware Preference Optimization (PNAPO), which augments each preference sample by retaining the winner/loser prior noises and the reward gap, turning the standard triplet into a richer tuple. The method then exploits the straight-line property of RF to estimate intermediate states via noise-image interpolation and derives an RF-consistent DPO-style objective. In addition, the paper introduces a dynamic regularization schedule based on reward gap and training progress. Experiments on FLUX.1-dev and SD3-Medium show improved preference/alignment metrics and substantially lower training cost than Diffusion-DPO.

**Compliance With Llm Reviewing Policy:**

Affirmed.

**Final Justification:**

I change my score since the rebuttal addressed my main concerns.

**Key Questions For Authors:**

1. How should the efficiency comparison be interpreted when including offline data construction?
The paper reports a large reduction in training cost relative to Diffusion-DPO. Would this conclusion still hold if the one-time cost of generating the noise-tracked preference dataset were included? A clear accounting would strengthen the practical claims.
2. Can the authors clarify the relationship to recent flow-specific post-training methods?
In particular, how should readers compare PNAPO to online flow-model alignment approaches such as Flow-GRPO?
3. How robust is the method across reward models or preference sources?
The current setup seems centered on HPSv2.1. Have the authors tried alternative reward models or smaller subsets of genuine human preference annotations to test whether PNAPO’s benefit is stable?

**Limitations:**

yes

**Strengths And Weaknesses:**

# Pros
- The paper identifies a real mismatch between standard preference data formats and the structure of rectified flow models. The argument that prior noise is part of trajectory identity in RF is intuitive and, in my view, one of the strongest aspects of the paper. This gives the method a clearer model-specific rationale than simply porting diffusion DPO objectives to flow models.
- The main idea—retaining the prior noise in the offline dataset and using RF-consistent interpolation—sounds simple, but it is not trivial. It leads to a cleaner trajectory estimator for RF than independent forward noising. I also find the dynamic regularization design reasonable, especially the idea of adapting update strength based on reward gap and training stage.
- The method targets an important regime: offline, RL-free post-training for strong T2I backbones. This is practically relevant because online RL for image generation is often expensive and engineering-heavy. The reported reduction in training cost relative to Diffusion-DPO is substantial, which strengthens the practical case for the method.
- The paper evaluates on two strong RF backbones, reports multiple automatic metrics, includes a user study, and provides ablations on the proposed components. The overall trend is consistent: the method appears to improve preference/alignment quality while being more efficient.
# Cons
- While the paper is well motivated, the method is still a relatively incremental extension of the DPO/preference-optimization line. The main novelty comes from adapting the data representation and surrogate objective to RF geometry, rather than introducing a fundamentally new optimization principle. I think this is still publishable if the empirical evidence is strong, but the paper should position its contribution carefully.
- The comparisons to Diffusion-DPO, SFT, IPO, and CaPO are useful, but the paper would be stronger with broader comparison to more recent preference optimization methods, especially those that address trajectory/step-level supervision or efficiency (e.g., SPO-like or inversion-based methods), and with a clearer discussion of how PNAPO relates to flow-specific post-training methods such as Flow-GRPO. I understand that Flow-GRPO is online RL rather than offline preference optimization, so it is not a strictly matched baseline, but it is still an important point of reference for RF post-training.
- The training labels are derived from HPSv2.1 rather than human preference annotations. This is reasonable for scalability, but it also means the method is primarily aligning to a reward-model-induced preference signal. The paper would benefit from a clearer discussion of how much of the gain may come from matching HPS-style signals rather than broader human preference.
- The user study supports the claims, but it seems relatively small in scale. Given that the paper is about preference optimization, stronger human evaluation would make the claims more convincing.
- The intuition behind the surrogate seems sound, but I would like a more careful explanation of the assumptions behind the “tighter” estimator claim and a clearer statement of what is guaranteed theoretically versus what is mainly empirical intuition.

---

> ### Author Rebuttal · Authors · 2026-03-30
>
> Thank you for highly recognizing the value of our study and helpful feedback!
>
> ---
>
> **W1:** *How should the paper position its contribution?*
>
> We view PNAPO not as a single trick but as a *systematic treatment* of preference alignment for RF, with three complementary contributions: (1) a noise-augmented sextuple data format that preserves trajectory identity; (2) a tighter surrogate objective derived from RF straightness with formal KL bounds (Eq. 12); and (3) a dual-dimensional dynamic regularization based on reward margins and training progress. The experiment results demonstrate substantial practical value. We will sharpen the positioning in the revised paper.
>
> ---
>
> **W2/Q2:** *How does PNAPO compare to other methods?*
>
> We have included an additional experimental comparison with Flow-GRPO (online RL) and InPO[1] (inversion based), all trained with HPSv2.1:
>
> Supp Table 1:  Win-rate comparison on HPDv2
>
> |SD3-M|PickScore|HPSv2.1|ImReward|Aesthetic|CLIP|
> |-|-|-|-|-|-|
> |PNAPO vs InPO|62.7%|65.1%|55.1%|58.8%|56.4%|
> |PNAPO vs Flow-GRPO|60.4%|66.8%|59.7%|65.9%|53.0%|
>
> Supp Table 2:  Total cost comparison (SD3-M, H800 GPUs)
>
> |SD3-M|H800 GPU-hours|
> |-|-|
> |Flow-GRPO|~1200 (training + online rollouts)|
> |InPO|~87.6 (training only)|
> |PNAPO|~44.8 (20.8 training, 24 dataset construction)|
>
> PNAPO outperforms both while being substantially more efficient.
>
> [1] InPO: Inversion Preference Optimization with Reparametrized DDIM for Efficient Diffusion Model Alignment. CVPR 2025
>
> ---
>
> **W3:** *How much gain comes from matching HPS signals versus human preferences?*
>
> This is an important concern. We highlight two pieces of evidence:
>
> (1) Although training uses only HPSv2.1 labels, Table 2 shows that **all held-out metrics** (PickScore, ImageReward, Aesthetic, CLIP) improve consistently. If the model were merely overfitting to HPSv2.1's idiosyncratic patterns, we would not expect gains on independently trained reward models.
>
> (2) We conducted an experiment on SDXL comparing DPO trained with original human annotations (Pick-a-Pic) vs. HPSv2.1-relabeled annotations, evaluated on HPDv2:
>
> Supp Table 3: Win-rate on HPDv2 (SDXL)
>
> |SDXL|PickScore|HPSv2.1|ImReward|Aesthetic|CLIP|
> |-|-|-|-|-|-|
> |DPO (HPSv2.1 label) vs DPO (Human label)|56.3%|60.8%|53.2%|58.4%|51.5%|
>
> The reward-model-relabeled data outperforms human-labeled data across all metrics, consistent with findings in noisy-label literature [2,3] that learned pseudo-labels can effectively denoise crowd-sourced annotations. **HPSv2.1 acts as a data cleanser** rather than introducing a narrow bias.
>
> [2] Self-training with noisy student improves imagenet classification. CVPR 2020.
>
> [3] Rethinking pretraining and self-training. NeurIPS 2025
>
> ---
>
> **W4:** Can the scale of the user study be expanded to make the claims more convincing?
>
> We have collected a larger-scale user study with **40 participants** evaluating **60 triplets each** (2,400 total responses):
>
> Supp Table 4:  Expanded User Study
>
> | | Overall Preference|Visual Attractiveness|Text Alignment|
> |-|-|-|-|
> |FLUX| 8.50% （68 votes）|8.25% (66 votes) |11.88% (95 votes)|
> |DPO-FLUX|22.25% (178 votes)|17.12% (137 votes)| 20.38% (163 votes)|
> |PNAPO-FLUX|69.25% (554 votes )|74.62% (597 votes)|67.75% (542 votes)|
>
> The results are consistent with and reinforce our original findings.
>
> ---
>
> **W5:** *What are the explicit assumptions and theoretical guarantees behind the "tighter" estimator claim?*
>
> We detail this in the main text (left col. lines 240-248, right col. lines 177-194, Eq. 12). The tighter estimator stems from utilizing the actual prior noise stored in the dataset rather than resampling. Crucially, the 'endpoint conditioning' in the left-hand KL divergence is the key to this inequality holding. Full detailed proofs are provided in Appendix C.
>
> ---
>
> **Q1:** Does the training efficiency hold when accounting for the one-time offline data construction cost?
>
> Yes!  We report total costs including dataset construction:
>
> Supp Table 5:  Total cost comparison for SD3-M
>
> |SD3-M|H800 GPU-hours|
> |-|-|
> |Flow-GRPO| ~1200 (training + online rollouts)|
> |Diffusion-DPO|~ 249.6 (training only)|
> |PNAPO|~44.8 (20.8 training + 24 dataset construction)|
>
> Supp Table 6:  Total cost comparison for FLUX
>
> |FLUX|H800 GPU-hours|
> |-|-|
> |Diffusion-DPO|~ 422.4 (training only)|
> |PNAPO| ~104.8 (35.2 training, 69.6 dataset construction) |
>
> Even with dataset construction, PNAPO maintains a **4–5× advantage** over Diffusion-DPO and **~26×** over Flow-GRPO.
>
> ---
>
> **Q3:**  *Is PNAPO robust across alternative reward models beyond HPSv2.1?*
>
> Table 6 in our paper shows that PNAPO yields gains when trained with various reward models (PickScore, HPSv2.1, ImageReward, Aesthetic, CLIP). HPSv2.1 achieves the best balance because its multi-dimensional scoring (aesthetics + text alignment) aligns well with PNAPO's optimization goals. This is consistent across held-out metrics, indicating robustness rather than overfitting to any single reward signal.

---

> > ### Author Rebuttal · Reviewer_qGiQ · 2026-04-02
> >
> > All my concerns are resolved.

---

> > > ### Author Response · Authors · 2026-04-03
> > >
> > > Dear Reviewer qGiQ,
> > >
> > > We sincerely appreciate your time and effort in reviewing our paper and engaging with our rebuttal. We are very glad that our responses have fully resolved your concerns, and we are deeply grateful for your support and the updated score!
> > >
> > > Best regards,
> > >
> > > Authors

---

### Official Review · Reviewer_59gx · 2026-03-12

**Soundness:** 2
**Presentation:** 2
**Significance:** 2
**Originality:** 3
**Overall Recommendation:** 4
**Confidence:** 4

**Summary:**

The paper studies a largely overlooked problem where DPO style offline preference optimization only operates in the final generated images without looking at the initial noise priors. The paper proposes a simple and effective method by storing the noise priors which are then used to generate intermediate states through interpolation. The paper further proposes a dynamic regularization strategy to stablize the  training. Extensive experiments demonstrate the effectiveness of the proposed method.

**Compliance With Llm Reviewing Policy:**

Affirmed.

**Final Justification:**

thank the authors for the response, I think the proposed method is something worth a broader audience and further study. I have raise my score.

**Key Questions For Authors:**

1. Does it really worth giving up other high quality available dataset to trade for training speed? Have the authors tried compare the results of PNAPO using self-generated dataset vs other models using different datasets with potentially higher quality?
2. How does the proposed method work on video models?

**Limitations:**

yes

**Strengths And Weaknesses:**

## Strength
1. The paper is well written and easy to follow with good motivation.
2. The paper studies a largely ignored problem by the community that the optimization process is less accurate when noise prior is not provided and proposes a simple method to improve upon baseline method.
3. Empirical results and theoretical derivation justifies the effectiveness of the proposed method

## Weakness
1. The method requires self-constructing training dataset which is resource-consuming and not viable.
2. The method relies on the straight-line property of RF which is not always guaranteed.
3. Due to the limitation mentioned in 2, there is no way to apply datasets that are generated by other models, greatly limiting the flexibility of offline optimization.
4. Since the proposed method requires pre-collecting dataset which is similar to online-RL with self-rollout, therefore online RL methods should be included for comparison under similar computation constraints.

---

> ### Author Rebuttal · Authors · 2026-03-30
>
> We're truly grateful for your insightful and thorough feedback!
>
> ---
>
> **W1:** *The method requires self-constructing training dataset which is resource-consuming and not viable.*
>
> We respectfully note that self-generated training data is now the prevailing paradigm in RL-based diffusion alignment (Flow-GRPO, Dance-GRPO, Diffusion-NFT). The key question is *total computational cost*, where PNAPO is highly competitive:
>
> Supp Table 1:  Total cost comparison for SD3-M (H800 GPUs)
>
> |SD3-M|H800 GPU-hours|
> |-|-|
> |Flow-GRPO|~1200 (training + online rollouts)|
> |Diffusion-DPO|~ 249.6 (training only) |
> |PNAPO|~44.8 (20.8 training + 24 dataset construction)|
>
> Supp Table 2:  Total cost comparison for FLUX (H800 GPUs)
>
> |FLUX|H800 GPU-hours|
> |-|-|
> |Diffusion-DPO|~422.4 (training only)|
> |PNAPO|~104.8 (35.2 training, 69.6 dataset construction)|
>
> Even including dataset construction, PNAPO remains **4–5× cheaper** than Diffusion-DPO and **~26× cheaper** than Flow-GRPO.
>
> ---
>
> **W2:** *The method relies on the straight-line property of RF which is not always guaranteed.*
>
> RF straightness is a statistical approximation rather than an absolute per-trajectory guarantee. The key comparison is between PNAPO's interpolation error and Diffusion-DPO's independent noise injection. PNAPO's interpolation uses both true endpoints $(x_{0},x_{T})$, so the approximation error is bounded by the trajectory's deviation from a straight line, which is small by RF's training objective when the prior noise is stored. In contrast, Diffusion-DPO uses noise that is completely independent of the actual generation trajectory, resulting in a substantially larger mismatch. This is formalized in Eq. 12, where we show the KL divergence of our approximation is provably no larger than Diffusion-DPO's. The ablation in Table 4 empirically confirms that "+prior noise" consistently improves over the independent-noise baseline.
>
> ---
>
> **W3:** *There is no way to apply datasets generated by other models, greatly limiting flexibility.*
>
> We acknowledge this constraint for the default setting. However, we offer two perspectives:
>
> 1. Self-improvement via self-generated data is an **established and principled** alignment strategy, shared with LLM self-play (SPIN) and online RL methods (Flow-GRPO). The model-specificity of data is inherent to all these approaches.
>
> 2. PNAPO **can** leverage external datasets via RF-inversion, which estimates the prior noise from the target model's own dynamics. We validated this by fine-tuning SD3-M on OPDv1 [1] using RF-inversion to estimate SD3-M-compatible noise:
>
>    Supp Table 3: Win-rate comparison on OPDv1:
>
>    |SD3-M|PickScore|HPSv2.1|ImReward |Aesthetic|CLIP|
>    |-|-|-|-|-|-|
>    |PNAPO vs DPO|63.9%|75.2%|60.1%|69.8%|64.1%|
>    |PNAPO vs IPO|62.4%|73.6%|59.7%|64.0%|63.8%|
>
>    [1] https://huggingface.co/datasets/data-is-better-together/open-image-preferences-v1
>
> ---
>
> **W4:** *Online RL methods should be included for comparison under similar computation constraints.*
>
> Excellent suggestion! We have added a comparison with Flow-GRPO, both trained with HPSv2.1:
>
> Supp Table 4:  Win-rate comparison on HPDv2 (SD3-M)
>
> |SD3-M|PickScore|HPSv2.1|ImReward|Aesthetic|CLIP|
> |-|-|-|-|-|-|
> |PNAPO vs Flow-GRPO|60.4%|66.8%|59.7%|65.9%|53.0%|
>
> PNAPO outperforms Flow-GRPO across most metrics while requiring ~26× less compute (Supp Table 1), validating the effectiveness of our offline approach.
>
> ---
>
> **Q1:** *Does it really worth giving up other high-quality available datasets?*
>
> Constructive suggestion! Our primary goal is to provide a general self-improvement framework, noting that the definition of a 'high-quality available dataset' is relative (e.g., OPDv1 is high-quality for SD3-M, but not for FLUX). Accordingly, we provide additional comparative results for fine-tuning SD3-M: 1) PNAPO (self-generated data) vs. DPO (OPDv1); and 2) PNAPO (self-generated data) vs. PNAPO (OPDv1 + RF inversion).
>
> Supp Table 5: Win-rate comparison on HPDv2:
>
> |SD3-M|PickScore|HPSv2.1|ImReward|Aesthetic|CLIP|
> |-|-|-|-|-|-|
> |PNAPO (self generated) vs DPO (OPDv1)|62.0%|66.3%|55.1%|58.2%|57.8%|
> |PNAPO (self generated) vs PNAPO (OPDv1+RF Inversion)|57.3%|59.4%|52.3%|56.5%|54.3%|
>
> ---
>
> **Q2:** *How does the proposed method work on video models?*
>
> We fine-tuned the Wan2.1-1.3B-T2V video generation model using both PNAPO and DPO, employing VideoReward [2] (VQ:MQ:TA = 1:1:1) for dataset scoring:
>
> Supp Table 6: VBench comparison
>
> |Wan2.1-1.3B-T2V|Total Score|Quality Score |Semantic Score|
> |-|-|-|-|
> |Pretained-Wan2.1|82.67|83.74|78.38|
> |DPO-Wan2.1|83.13|83.86|80.19|
> |PNAPO-Wan2.1|**83.55**|**84.12**|**81.26**|
>
> [2] Improving Video Generation with Human Feedback. NeurIPS 2025

---

> > ### Author Rebuttal · Reviewer_59gx · 2026-04-04
> >
> > Thank the authors for the response, I read through other reviewers' comments. I still think there is more to study regarding the proposed method such as the comprehensiveness of the experiments and training dynamics. But I will reconsider my score.

---

> > > ### Author Response · Authors · 2026-04-04
> > >
> > > Dear Reviewer 59gx,
> > >
> > > Thank you very much for taking the time to review our rebuttal and for your willingness to reconsider the score! We truly
> > > appreciate your continued engagement with our work. Below, we do our
> > > best to address the two points you raised.
> > >
> > > **1. Training Dynamics**
> > >
> > > We would like to respectfully draw your attention to several analyses
> > > in our paper and rebuttal that we believe speak directly to the
> > > training dynamics of PNAPO:
> > >
> > > (a) **Gradient-level analysis** (Section 4.3, Eq. 13): We analytically
> > > decompose how $\beta$ and the preference margin jointly govern gradient
> > > magnitude, providing insight into why a fixed $\beta$ becomes suboptimal as
> > > training progresses.
> > >
> > > (b) **Orthogonal roles of f(δr) and g(n)** (Table 5; also discussed in
> > > detail in our response to Reviewer j33J): We demonstrate that $f(\delta r)$
> > > and $g(n)$ address two complementary aspects of training dynamics:
> > > $f(\delta r)$ improves *learning efficiency* by upweighting high-confidence
> > > preference pairs, while $g(n)$ ensures *training stability* by
> > > gradually reducing the KL penalty strength in later stages. Their
> > > combination $\beta(\delta r, n)$ achieves the best performance across all metrics
> > > (Table 5), confirming their complementary nature.
> > >
> > > (c) **Training schedule sensitivity** (Table 7): We evaluate five
> > > different $(n_1, n_2)$ annealing configurations, characterizing how the
> > > schedule affects final performance.
> > >
> > > (d) **Mechanistic justification** (Section 4.3, Lines 252–261;
> > > expanded in our response to Reviewer j33J): We explain why stronger
> > > regularization early in training drives meaningful parameter updates,
> > > and why cosine decay of $g(n)$ is necessary to prevent
> > > "alignment-forgetting" oscillation in later stages. We also provide a
> > > curriculum-learning interpretation of $f(\delta r)$'s sigmoid design.
> > >
> > > We note that Reviewer j33J raised closely related questions regarding
> > > training dynamics and the design rationale of $f$ and $g$, and kindly
> > > confirmed that our analyses fully resolved their concerns.
> > >
> > > **2. Comprehensiveness of Experiments**
> > >
> > > We are grateful for your high standards and share the goal of making
> > > the evaluation as thorough as possible. Across the original paper and
> > > our rebuttal responses to all reviewers, our evaluation now covers:
> > >
> > > - 3 model architectures: FLUX, SD3-M (image) + Wan2.1-1.3B (video)
> > > - 4 text-to-image benchmarks (HPDv2, OPDv1, Diffusion-DB, GenEval) + VBench
> > >   for video
> > > - 6 evaluation metrics: PickScore, HPSv2.1, ImageReward, Aesthetic,
> > >   CLIP, and VisualQuality-R1
> > > - 6 baselines: DPO, SFT, IPO, CaPO, Flow-GRPO (online RL), InPO
> > >   (inversion-based)
> > > - External dataset experiment via RF-inversion on OPDv1
> > >   (Supp Tables 3 & 5)
> > > - Self-generated vs. external data comparison (Supp Table 5)
> > > - 4 ablation studies: proposed improvement, dynamic regularization components,
> > >   reward model robustness, and hyperparameter sensitivity
> > > - User study expanded to 40 participants / 2,400 responses
> > > - Full computational cost analysis including dataset construction
> > > - Cross-label-source comparison: HPSv2.1-labeled vs. human-labeled
> > >   data (Supp Table 3 in our response to Reviewer qGiQ)
> > >
> > > We have made every effort to be comprehensive, and we hope the scope
> > > above reflects that. That said, we deeply respect your expertise and
> > > would be sincerely grateful if you could share any specific
> > > experiments or analyses that you feel would further strengthen the
> > > paper. Concrete suggestions would be invaluable in helping us
> > > prioritize the most meaningful improvements for our paper.
> > >
> > > Thank you once again for your thoughtful and constructive feedback!
> > > We look forward to your response.
> > >
> > > Best regards,
> > >
> > > Authors

---

### Official Review · Reviewer_2duZ · 2026-03-13

**Soundness:** 3
**Presentation:** 3
**Significance:** 3
**Originality:** 3
**Overall Recommendation:** 4
**Confidence:** 4

**Summary:**

This paper introduces a method called PNAPO (Prior Noise-Aware Preference Optimization) , an offline preference alignment framework, which is specifically designed to enhance the performance of text to image generation models based on Rectified Flow.

The current T2I model preference optimization (such as DPO method) usually only saves the final generated "winning" and "losing" images, while discarding the initial noise used to generate these images. The existing methods often estimate the generated trajectory through an independent forward denoising process, which not only does not match the real inverse denoising dynamics, but also introduces unnecessary variance and high computational costs. PNAPO uses a dataset that preserves prior noise, interpolation based trajectory estimation, and dynamic regularization strategies to address these issues. Through experiments, it has been proven that PNAPO consistently and significantly outperforms existing Diffusion DPO and other baseline models in human evaluation and quantitative metrics such as text alignment, visual aesthetics, and realism, while significantly reducing computational costs.

**Compliance With Llm Reviewing Policy:**

Affirmed.

**Final Justification:**

The authors have given the results of applying proposed methods on existing Preference dataset in the rebuttal stage, which is my biggest concern. Overall the proposed methods is interesting and sound, demonstrate better performance than dpo, so overall I will give the paper a weak accept score.

**Key Questions For Authors:**

1. If the dataset only records one xT, when the model's parameters shift during alignment, the original xT may no longer correspond to the current x0. Will this 'outdated' trajectory information mislead subsequent optimization?
2. Figure 4, Four cats surrounding a dog. In the images generated by PNAPO-SD3, the dog has obvious cat features. Can the author explain this phenomenon?
3. Refer to weaknesses section.

**Limitations:**

Yes.

**Strengths And Weaknesses:**

Strengths:

1. Improvement in generation performance: PNAPO consistently and significantly outperforms existing Diffusion DPO and other baseline models in human evaluation and quantitative metrics such as text alignment, visual aesthetics, realism, etc.
2. Reduction in computational costs: PNAPO significantly reduces computational costs.

Weaknesses

1. Currently, mainstream large-scale preference datasets only store images and do not contain noise. Does this mean that PNAPO cannot utilize these existing datasets?
2. There are only 10 users in the User Study, which is a small sample size.
3. The dataset lacks cross model generalization ability.
4. Although the author explained the reasons for not comparing with online methods, considering the limitations of the author's dataset's generalization ability and the use of reward models in dataset construction, it seems necessary to compare with online methods.

---

> ### Author Rebuttal · Authors · 2026-03-30
>
> We sincerely thank you for your valuable and constructive suggestions!
>
> ---
> **W1:** *Can PNAPO utilize existing image-only preference datasets that lack noise data?*
> Yes, PNAPO can be readily adapted to existing image-only datasets. While **our primary design retains the actual prior noise for tighter trajectory estimation**, for datasets without noise, we can employ RF-inversion [1] to recover an approximate prior noise for each image from the target model's perspective. This is a one-time, parallelizable preprocessing step. We validated this by fine-tuning SD3-M on the OPDv1 [2] dataset using RF-inversion to estimate prior noise. We report win-rates on the HPDv2 test set:
>
> Supp Table 1: Win-rate on HPDv2 (RF-inversion for PNAPO)
>
> |SD3-M|PickScore|HPSv2.1|ImReward|Aesthetic|CLIP|
> |-|-|-|-|-|-|
> |PNAPO vs DPO|63.9%|75.2%| 60.1% |69.8%|64.1%|
> |PNAPO vs IPO |62.4%|73.6%| 59.7% |64.0%|63.8%|
>
> Importantly, existing public datasets are limited in both quantity (e.g., OPDv1 contains ~8.7k pairs primarily from SD3.5-L and FLUX-Dev) and quality (e.g., Pick-a-Pic mainly contains data from outdated models), yielding minimal gains for new, high-quality models. Self-generated data for self-improvement has become a mainstream paradigm, as adopted by Flow-GRPO, Dance-GRPO, and LLM self-play methods.
>
> [1] Semantic Image Inversion and Editing using Rectified Stochastic Differential Equations. ICLR 2025
>
> [2] https://huggingface.co/datasets/data-is-better-together/open-image-preferences-v1
>
> ---
> **W2:** *There are only 10 users in the User Study, which is a small sample size.*
>
> We have conducted a larger-scale user study with **40 participants**, each evaluating **60 randomly selected image triplets** (FLUX / DPO-FLUX / PNAPO-FLUX), totaling **2,400 responses**. Results confirm our original findings:
>
> Supp Table 2:  Expanded User Study
>
> | | Overall Preference|Visual Attractiveness|Text Alignment|
> |-|-|-|-|
> | FLUX | 8.50% (68 votes)| 8.25% (66 votes) | 11.88% (95 votes) |
> | DPO-FLUX| 22.25% (178 votes) | 17.12% (137 votes) | 20.38% (163 votes) |
> | PNAPO-FLUX | 69.25% (554 votes ) | 74.62% (597 votes)| 67.75% (542 votes) |
>
> ---
>
> **W3:** *The dataset lacks cross model generalization ability.*
>
> We acknowledge that our self-generated dataset is model-specific by design. This is consistent with the self-improvement paradigm widely adopted in RL-based alignment, such as Flow-GRPO (on policy). The rationale is that aligning a model with its own generated data maximizes training stability by avoiding distribution mismatch. When aligning a different model, PNAPO (off policy) simply generates a new dataset using that model. That said, as demonstrated in **W1**, PNAPO can also leverage external datasets via RF-inversion, providing flexibility when model-specific data generation is impractical.
>
> ---
>
> **W4:** *It seems necessary to compare with online methods.*
>
> We appreciate this suggestion and have added a comparison with Flow-GRPO. Using HPSv2.1 as the reward model for both methods:
>
> Supp Table 3:  Win-rate comparison on HPDv2 (SD3-M):
>
> |SD3-M| PickScore | HPSv2.1 | ImReward | Aesthetic | CLIP |
> |-|-|-|-|-|-|
> |PNAPO vs Flow-GRPO|60.4%|66.8%|59.7%|65.9%|53.0%|
>
> Supp Table 4:  Total computational cost comparison (SD3-M, H800 GPUs):
>
> |SD3-M|H800 GPU-hours|
> |-|-|
> |Flow-GRPO| ~1200|
> |PNAPO| ~44.8 (20.8 for training, 24 for dataset construction)|
>
> PNAPO achieves superior alignment with ~26× less total compute, demonstrating the efficiency advantages of our offline framework.
>
> ---
>
> **Q1:** *Will this 'outdated' trajectory information mislead subsequent optimization?*
>
> This is an insightful question about distributional shift in off-policy learning. We argue the stored trajectories remain valid for three reasons:
>
> (1) DPO optimizes log-ratios, not absolute likelihoods. The objective (Eq. 10) compares the *relative* velocity prediction errors of the policy and reference models along a fixed trajectory. Even as parameters shift, the optimization signal remains meaningful.
>
> (2) The stored ($x_{0}$, $x_{T}$) pair defines a trajectory that is intrinsic to the data manifold, not to the model's current parameters. The model learns to better predict the velocity field along trajectories that lead to preferred outcomes. This is directly analogous to standard LLM-DPO, where prompts and completions are fixed while model parameters update.
>
> ---
>
> **Q2:** *Figure 4, Four cats surrounding a dog. In the images generated by PNAPO-SD3, the dog has obvious cat features.*
>
> We attribute this to a limitation of HPS v2.1: it primarily evaluates holistic aesthetic quality but is relatively insensitive to fine-grained compositional correctness (e.g., each animal retaining distinct category features). During optimization, the model trades marginal compositional accuracy for aesthetic attributes that HPS v2.1 rewards more heavily. This tendency is not specific to PNAPO, any RL method (incorporates HPS v2.1 specific scores) would face the same issue.

---

> > ### Author Rebuttal · Reviewer_2duZ · 2026-04-02
> >
> > I think the authors have beed address all of my concerns, i have no further questions, and will raise my score.

---

> > > ### Author Response · Authors · 2026-04-02
> > >
> > > Dear Reviewer 2duZ,
> > >
> > > Thank you very much for your time, your constructive feedback, and your positive acknowledgment of our rebuttal. We are thrilled to hear that our responses have fully addressed all of your concerns.
> > >
> > > We are deeply encouraged by your positive response and the updated score.  Please rest assured that we will carefully incorporate all of your valuable suggestions into the final version of our paper.
> > >
> > > Thank you once again for your invaluable guidance and support in helping us improve our paper!
> > >
> > > Best regards,
> > >
> > > Authors

---

### Decision · Program_Chairs · 2026-04-30

**Decision:**

Accept (regular)

**Comment:**

In this paper, authors proposed prior noise-aware preference optimization (PNAPO), an off-policy alignment framework for rectified flow. Authors leveraged the straight-line property of RF and estimated intermediate states via noise-image interpolation and yielded a tigher surrogate objective for preference optimization. Experimental results shows the effectiveness of the proposed methods.

The final rating of this paper is 1 Accept and 3 weak accept.

Before rebuttal, Reviewers thoughts

The strength of the paper are:

1) Improvement in generation performance. (Reviewer 2duZ)
2) Reduction in computational costs. (Reviewer 2duZ)
3) well written and easy to follow with good motivation. (Reviewer 59gx, j33J)
4)  studies a largely ignored problem. (Reviewer 59gx)
5) Empirical results and theoretical derivation justifies the effectiveness of the proposed method. (Reviewer 59gx)
6)  identifies a real mismatch between standard preference data formats and the structure of rectified flow models. (Reviewer qGiQ)
7) The main idea sounds simple, but it is not trivial. (Reviewer qGiQ)
8) targets an important regime. (Reviewer qGiQ)
9) comprehensive results. (Reviewer qGiQ, j33J)

weaknesses are:
1) utilize existing datasets? (Reviewer 2duZ)
2) only 10 users in the User Study. (Reviewer 2duZ)
3) dataset lacks cross model generalization ability. (Reviewer 2duZ)
4) compare with online methods. (Reviewer 2duZ)
5) requires self-constructing training dataset which is resource-consuming and not viable. (Reviewer 59gx)
6) relies on the straight-line property of RF which is not always guaranteed. (Reviewer 59gx)
7) no way to apply datasets that are generated by other models. (Reviewer 59gx)
8) online RL methods should be included for comparison under similar computation constraints. (Reviewer 59gx)
9) relatively incremental extension of the DPO/preference-optimization line. (Reviewer qGiQ)
10) need to compare with more methods.  (Reviewer qGiQ)
11) clearer discussion of how much of the gain may come from matching HPS-style signals rather than broader human preference. (Reviewer qGiQ)
12)  user study seems relatively small in scale. (Reviewer qGiQ)
13) more careful explanation of the assumptions behind the “tighter” estimator claim. (Reviewer qGiQ)
14) presentation of the paper needs significant improvement. (Reviewer j33J)
15) several arguments are not entirely convincing. (Reviewer j33J)
16) include additional visual quality metrics. (Reviewer j33J)

After rebuttal,
Reviewer 2duZ thought their biggest concerns is addressed and gave weak accept rating.

Reviewer 59gx raised rating to weak accept.

Reviewer qGiQ mentioned their concerns are addressed and changed score to accept.

Reviewer j33J thought the paper is highly insightful and gave a weak accept.

Given all these AC decided to accept this paper.